# Uncovering Bias Mechanisms in Observational Studies

Ilker Demirel [* 1]   Zeshan Hussain [* 1 2]   Piersilvio De Bartolomeis [3]   David Sontag [1]

## Abstract

Observational studies are a key resource for causal inference but are often affected by systematic biases. Prior work has focused mainly on detecting these biases, via sensitivity analyses and comparisons with randomized controlled trials, or mitigating them through debiasing techniques. However, there remains a lack of methodology for uncovering the underlying mechanisms driving these biases, *e.g.*, whether due to hidden confounding or selection of participants. In this work, we show that the relationship between bias magnitude and the predictive performance of nuisance function estimators (in the observational study) can help distinguish common sources of bias. We validate our methodology through extensive synthetic experiments and a real-world case study, demonstrating its effectiveness in revealing the mechanisms behind observed biases. Our framework offers a new lens for understanding and characterizing bias in observational studies, with practical implications for improving causal inference.

## 1. Introduction

Observational data from electronic health records and insurance claims offer compelling advantages for causal inference. They incur less cost to collect and provide broader population representation than randomized controlled trials (RCT), allowing analyses in subgroups not supported in RCTs (GOVC, 2019; NICE, 2022). However, observational studies (OS) are plagued by biases that undermine their reliability, such as hidden confounding or selection bias from loss to follow-up (Hernán et al., 2008; Imbens & Rubin, 2015; Pearl & Mackenzie, 2018; Lodi et al., 2019).

It is crucial for practitioners to have a nuanced understanding of the bias within an OS to improve the study design. A good example is Dagan et al. (2021), where the authors iteratively refine their covariate adjustment set via calibration with negative controls to estimate the efficacy of COVID-19 vaccinations (see Section 3). To tackle this issue, we develop methodology to distinguish between a set of causal bias mechanisms commonly found in OSes.

Our approach is related to two lines of work[1 2]: i) benchmarking OSes against RCTs (Hartman et al., 2015; Kallus et al., 2018; Forbes & Dahabreh, 2020; Hussain et al., 2022; Demirel et al., 2024b), and ii) recovering a causal graph structure (Shimizu et al., 2006; Shanmugam et al., 2015; Heinze-Deml et al., 2018; Glymour et al., 2019; Vowels et al., 2022; Choo et al., 2022; Squires & Uhler, 2023). While the benchmarking literature focuses on spotting biases, it does not provide insights as to the underlying reasons. The structure learning literature attempts to retrieve a causal graph from observed data, which would describe the exact bias mechanism. However, it requires strong assumptions which are unrealistic in practice.

Instead of recovering an exact graph, we distinguish between families of causal graphs corresponding to different bias mechanisms (see Section 3), affording practitioners actionable insights to improve their study design. This task is more tractable, as it groups together several graphs that exhibit equivalent bias patterns. To that end, we first estimate the bias function in the OS using data from an RCT conducted on a population represented in the OS. We then examine how bias varies across patients as a function of the predictive performance of the *nuisance function estimators* in the OS. This relationship gives rise to statistics that can characterize distinct bias mechanisms.

**Contributions**   First, we establish a comprehensive taxonomy of common causal biases, each corresponding to a distinct class of graphs (Section 3). We describe a generative model for the OS in the context of these graphs motivated by clinical decision-making (Section 4.1). Next, we demonstrate a relationship between the predictive performance of the nuisance functions and the bias function in the OS (Section 4.2). To quantify this relationship, we use the covariance between the prediction error and magnitude of the

---

[*]Equal contribution  [1]MIT [2]Brigham and Women's Hospital and Harvard Medical School [3]ETH Zurich. Correspondence to: Ilker Demirel <demirel@mit.edu>.

*Proceedings of the 43rd International Conference on Machine Learning*, Seoul, South Korea. PMLR 306, 2026. Copyright 2026 by the author(s).

---

[1]An extended related work section is in Section C.
[2]Codebase: https://github.com/demireal/benchmarking-os

bias, which enables provable discrimination of different bias mechanisms (Section 4.3). For practical implementation, we propose consistent estimators of the covariance (Section 4.5). Finally, we conduct both synthetic experiments and real-world analysis using data from Women's Health Initiative (TWHI, 1998) (Sections 5 and 6).

## 2. Notation and Background

Let $A$ denote a treatment action, $X$ the set of *measured* set of patient covariates, and $Y$ the outcome of interest. We denote by $Y^a$ the *potential* outcome under the treatment action $A = a$. For each patient $i$, we observe only one of their *potential* outcomes, that is, $Y_i = Y^{A_i}$.

We assume access to patient-level data from an RCT and an OS, and use $R = 1$ and $R = 0$ to represent the underlying RCT and OS *populations*, respectively. We use $S$ to denote whether a patient was *selected* into the study cohort for analysis. For instance, a patient may be excluded from the analysis in an RCT if they did not adhere to their treatment assignment (*i.e.,* $R_i = 1$ and $S_i = 0$). From a large insurance claims dataset, a patient may be selected into the OS cohort when emulating an RCT if they meet the eligibility criteria (*i.e.,* $R_i = 0$ and $S_i = 1$). We assume that $X$ is available for all patients, and that $A$ and $Y$ are available for those selected into the analysis ($S_i = 1$).

Finally, we let $U$ denote the set of *unmeasured* covariates that can influence the downstream variables $S$, $A$, and $Y^a$. Such omitted variables are the reasons behind common causal biases in OSes, which are described in Section 3.

The causal estimand we focus on is the conditional average treatment effect (CATE), defined as

$$\text{CATE}_{\text{rct}}(x, a) := \mathbb{E}[Y^a \mid X = x, R = 1]. \quad (1)$$
$$\text{CATE}_{\text{os}}(x, a) := \mathbb{E}[Y^a \mid X = x, R = 0]. \quad (2)$$

We use the term CATE slightly differently from the conventional contrast-based definition. Traditionally, the conditional average treatment effect is defined as $\mathbb{E}[Y^1 - Y^0 \mid X, R]$. In this work, we instead refer to the action-specific conditional potential-outcome means $\mathbb{E}[Y^a \mid X, R]$ as CATE functions indexed by the treatment action $a$. This is for analytical convenience: our covariance signatures are derived separately for each action $a$, and the usual contrast-based CATE can be recovered by taking the difference between the corresponding action-specific functions.

Estimating the CATE function is challenging when unobserved covariates, $U$, influence the treatment assignment ($A$), outcome ($Y$), and selection into the study cohort ($S$). We list below necessary conditions to identify the CATE function in a population, which are often satisfied in RCTs, but can be violated in OSes.

**Assumption 2.1** (Internal validity of RCT). The following hold in the RCT ($R = 1$) for all $a \in \{0, 1\}$.

*Ignorability of selection –* $Y^a \perp\!\!\!\perp S \mid X, R = 1$.
*Ignorability of treatment –* $Y^a \perp\!\!\!\perp A \mid X, S, R = 1$.
*Positivity –* $P(S = 1, A = a \mid X, R = 1) > 0$.

Under Assumption 2.1, $\text{CATE}_{\text{rct}}(x, a)$ is identified with

$$g_a(X) := \mathbb{E}[Y \mid X, R = 1, S = 1, A = a] \quad (3)$$
$$= \text{CATE}_{\text{rct}}(x, a),$$

which can be estimated from the RCT. The outcome model in the OS population is defined similarly below.

$$f_a(X) := \mathbb{E}[Y \mid X, R = 0, S = 1, A = a], \quad (4)$$

which in general is not a valid identification of $\text{CATE}_{\text{os}}(x, a)$. We define the bias function in the OS,

$$b_a(x) := g_a(x) - f_a(x) = \underbrace{\text{CATE}_{\text{os}}(x, a) - f_a(x)}_{\text{Internal bias in the OS}}$$
$$+ \underbrace{\text{CATE}_{\text{rct}}(x, a) - \text{CATE}_{\text{os}}(x, a)}_{\text{Transportability bias}}. \quad (5)$$

There could be various mechanisms underlying the bias, which cannot be inferred from $b_a(x)$ alone. We develop methodology to uncover which mechanism could be driving the bias. To that end, we first give a taxonomy of common bias mechanisms in the next section.

## 3. A Taxonomy of Biases

We bucket the causal bias mechanisms into two categories. The first bucket involves an *unobserved* covariate affecting downstream variables. The second category is "collider" bias, where conditioning on a downstream variable induces spurious associations between the treatment and outcome.

### 3.1. Category 1: Unobserved Covariates

**Transportability Bias** Transportability bias arises when an unobserved covariate $U$ affects outcomes $Y^a$ and has different distributions across the OS and RCT populations (Figure 1a), even if the OS has internal validity, *i.e.,* $f_a(X) = \text{CATE}_{\text{os}}(x, a)$. Observe that

$$P(Y^a|X, R) = \sum_u P(Y^a|X, U = u, R)P(U = u|X, R).$$

When $P(U|X, R = 0) \neq P(U|X, R = 1)$, we have $\text{CATE}_{\text{os}}(x, a) \neq \text{CATE}_{\text{rct}}(x, a)$, hence $b_a(X) \neq 0$ due to Eq. (5). For example, age could be an effect modifier, whereby a chemotherapy for breast cancer has a larger effect on younger women (studied in RCT) compared to older women (studied in OS) (Ring et al., 2021). If age is not adjusted for, transportability bias will occur.

$P(U|R{=}0) \neq P(U|R{=}1)$

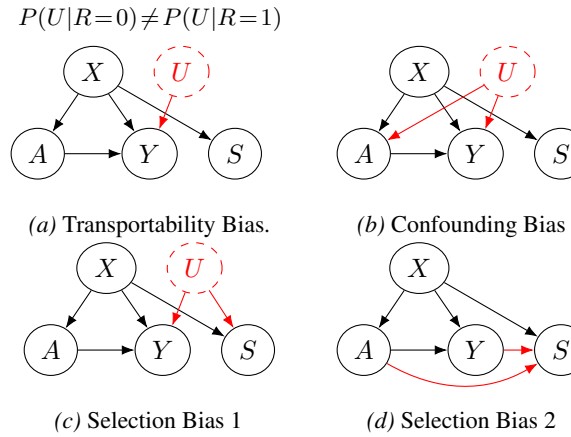

*(a)* Transportability Bias.  *(b)* Confounding Bias

*(c)* Selection Bias 1  *(d)* Selection Bias 2

*Figure 1.* Graphs of common biases in observational studies. $U$ denotes unmeasured covariates.

**Confounding Bias**  Confounding bias is a core impediment to causal inference in an OS. When covariates $X$ miss factors ($U$) that affect both the treatment assignment $A$ and outcomes $Y$ (Figure 1b), the "ignorability of treatment" condition in Assumption 2.1 is violated in OS ($R = 0$). This violation implies $f_a(X) \neq \text{CATE}_{\text{os}}(x, a)$, leading to non-zero bias $b_a(X)$.

To illustrate, consider COVID-19 vaccination studies: patients with higher socioeconomic status (SES) may be more likely to both receive vaccines and implement preventive health precautions. When an OS inadequately captures SES variations ($U$), we risk introducing confounding bias that inflates vaccine effectiveness estimates (Dagan et al., 2021).

**Selection Bias Type 1**  Selection bias emerges when the inclusion criteria lead to a cohort that does not represent the entire population. We examine two graphs capturing a diverse set of selection mechanisms. The first type (Figure 1c) arises when an unmeasured covariate $U$ affects selection $S$ and outcome $Y$, violating "ignorability of selection" in Assumption 2.1 in the OS ($R{=}0$).

Excluding patients who were lost-to-follow-up from the cohort can lead to this bias. As an example, Hernán et al. (2004) give antiretroviral therapy for preventing AIDS in HIV-infected patients, where true unmeasured immunosuppression level, $U$, affects both the AIDS risk ($Y$), and follow-up ($S$) through adverse side effects.

As another example, consider studies where participants opt-in to share their health data via a smart watch. Individuals with healthier lifestyles, such as a balanced diet ($U$) may be more likely to participate ($S = 1$). Since these individuals typically experience lower heart failure (HF) rates, $Y$, the study may underestimate the interventions' benefits (*e.g.*, regular jogging) for reducing HF rates in wider population.

### 3.2. Category 2: Conditioning on Colliders

**Selection Bias Type 2**  This bias (Figure 1d) arises when selection is affected by the treatment assignment and outcome, and it is particularly elusive (Holmberg & Andersen, 2022; Lipsky & Greenland, 2022). Initial analyses of the Women's Health Initiative OS showed protective effects of post-menopausal combined hormonal therapy against coronary heart disease and stroke (Barrett-Connor & Grady, 1998). However, subsequent analysis of the RCT component revealed increased risk (Manson et al., 2003). The discrepancy stemmed from selective enrollment: patients who initiated treatment pre-enrollment and continued without early adverse events were preferentially selected ($A \to S$), while early events precluded selection into the treatment group ($Y \to S$), deflating event rates in the treatment cohort (Prentice et al., 2005).

## 4. Identifying the Bias Type via Alignment with Predictive Performance

Here we present our methodology to distinguish between the bias mechanisms in Figure 1. We begin with a clinical example to motivate the data-generating process and lay out key assumptions in Section 4.1. In Section 4.2, we identify statistics that uniquely characterize bias mechanisms and derive guarantees in Sections 4.3 and 4.4. In Section 4.5, we develop consistent estimators of those statistics.

The next section introduces clinically-grounded data-generating assumptions that facilitate sharp theoretical analysis (Theorems 4.4, 4.6). However, our methods' practical utility and core insights from our analytical results extend beyond these assumptions. Table 1 provides a list of the assumptions with their role in our analysis and how much they can be relaxed in practice.

For example, we assume a single binary $U$ variable for theoretical convenience. However, we run synthetic experiments where $U$ is continuous and show that the results closely align with our theoretical predictions (see Section B.1). Similarly, our real-world case study in Section 6 involves multiple, continuous unmeasured confounders, yet our methodology provides clinically meaningful insights. Further, while we assume that only one of the biases in Figure 1 is at play for theoretical analysis, we provide experiments and interpretations under multiple bias types in Sections 6 and B.1.

### 4.1. Clinically Motivated Generative Model

We consider a binary treatment $A$, outcome $Y$, unmeasured covariate $U$, and a general covariate set $X \in \mathcal{X}$ which can include both categorical and continuous features. The generative model for downstream variables is given in Algorithm 1. Effectively, $U$-bias =False for

**Algorithm 1** Generative Model in the OS

**Input:** $T \in \{S, A, Y^0, Y^1\}$; Boolean $U$-bias;
Bernoulli Parameter Distribution $\mathcal{F}(p)$.

---

Let $p_{x,u}^T \coloneqq P(T = 1 \mid X = x, U = u, R = 0)$
**for** $x \in \mathcal{X}$ **do**
   $p_{x,u=0}^T \sim \mathcal{F}(p)$
   **if** $U$-bias **then**
      $p_{x,u=1}^T \sim \mathcal{F}(p)$
   **else**
      $p_{x,u=1}^T = p_{x,u=0}^T$
   **end if**
**end for**

---

$T \in \{S, A, Y^0, Y^1\}$ means that $U$ has no residual effect on $T$ after conditioning on $X$, and vice versa if $U$-bias $=$`True`.

**Magnitude of $U$'s effect varies across $X$.** The influence of an unmeasured covariate $U$ on a downstream variable $T \in \{S, A, Y^0, Y^1\}$ may vary across patients $X = x$. Specifically, its effect is *small* when $p_{x,u=1}^T \approx p_{x,u=0}^T$, and large otherwise (see Algorithm 1). This property forms the foundation of our methods as we discuss in Section 4.2, which we illustrate with an example from clinical practice.

**Clinical reasoning induces *contextual* independencies.** When a patient presents, the clinician follows a systematic approach, beginning with their history and symptoms, followed by diagnostic tests. This process involves evaluating clinical findings and test results. Based on these initial findings, distinct diagnostic pathways emerge, where the importance of subsequent variables may change dramatically.

For example, consider the evaluation of thyroid nodules. The initial assessment includes demographics, medical history, physical examination, and thyroid function tests. In a young male patient with normal thyroid function and a solitary thyroid nodule ($X$), the family history ($U$) of thyroid cancer is critical in guiding immediate fine-needle aspiration (FNA), while menopausal status is irrelevant. Conversely, in a post-menopausal female patient with the same nodule but a history of radiation exposure ($X$), the family history ($U$), becomes relatively less influential in the diagnosis, as they already are in the high-risk group requiring a biopsy.

**Conditioning induces low uncertainty.** The vignette above reveals a key property of clinical decision-making: given more context for a patient, downstream uncertainty decreases. In our example, probability of getting a FNA treatment ($A$) is high for the male patient (given observed context $X$). That is, variance in management is low. Whereas for the female patient, uncertainty in treatment decision ($A$) may be higher due to the profound impact of unobserved

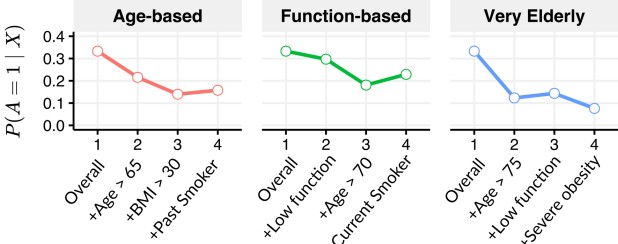

*Figure 2.* Treatment assignment probabilities in three subgroups in the WHI (TWHI, 1998). As the conditioning set is expanded and patient context is better specified, uncertainty in the decisions tend to decrease.

family history ($U$). Figure 2 illustrates this in real-world data, and we present detailed analyses in Sections 6 and B.2.

Thus, to model the "informed" nature of clinical decision-making, we focus on the scenarios where $P(T = 1 \mid X = x, U = u, R = 0)$ in Algorithm 1 is *away* from 0.5. To that end, we first define,

$$\mathcal{F}(p) \coloneqq \texttt{Unif}\big([0.1, p] \cup [1 - p, 0.9]\big), \ p \in (0.1, 0.5]. \quad (6)$$

which is a uniform distribution over the union of a *low-probability* and a *high-probability* range. This ensures that, given $X$ and $U$, the *uncertainty* in downstream variables $T \in \{S, A, Y^0, Y^1\}$ is low.[3] Next, we define the following *family* of distributions

$$\mathcal{F} \coloneqq \{\mathcal{F}(p) \mid p \in (0.1, 0.5]\}. \quad (7)$$

Note that $\mathcal{F}$ is quite rich and also includes `Uniform`$([0.1, 0.9])$ for $p = 0.5$ (*i.e.*, the whole interval). We fix 0.1 and 0.9 as lower and upper bounds to ensure *positivity*, which is a common and central assumption in causal inference. Finally, we state additional assumptions of our setting, which are also reflected in Figures 1a-1c (Figure 1d is studied in Section 4.4).

**Assumption 4.1.** For all $r, s, a, u \in \{0, 1\}$, we have

*Exogeneity of unmeasured covariates* – $X \perp\!\!\!\perp U \mid R$.
*Weak transportability* – $Y^a \perp\!\!\!\perp R \mid X, U$.
*Weak ignorability* – $Y^a \perp\!\!\!\perp S \perp\!\!\!\perp A \mid X, U, R$.
*Positivity* – $P(R = r, S = s, A = a, U = u \mid X) > 0$.

The prefix "weak" emphasizes the additional conditioning on the unmeasured covariate $U$.

### 4.2. Downstream Variance Drives the Bias

**DAGs miss contextual independencies.** While DAGs are useful it to depict causal biases, they paint an incomplete

---

[3]Entropy of a Bernoulli random variable $B$, which can be thought of as a measure of its "uncertainty", is maximized at $P(B = 1) = 0.5$, and shrinks toward 0 or 1.

Table 1. Summary of assumptions with their analytical role and practical implications.

| Assumption | Role in our Analysis | Relaxable in Practice? |
|---|---|---|
| Assumption 2.1 (RCT internal validity) | Ensures that RCT is a reliable source to estimate true effects. | **No.** If RCT is biased, the relative bias in OS cannot be disentangled. |
| $P(X = x|R = r) > 0$, $x \in \mathcal{X}, r \in \{0, 1\}$. | Limit analyses to overlapping patient population in RCT and OS. | **No.** For patient types that never appear in one of the cohorts, the bias ($b(X)$) cannot be estimated. |
| Consistency of Nuisance Estimators (Eq. 15-18) | Required for our estimators' consistency and asymptotical unbiasedness (Theorem 4.8). | **No.** Inconsistency (*e.g.,* model misspecification) introduces additional statistical bias on top of the causal bias, which then cannot be disentangled. |
| Binary & Univariate $U, S$ | Simple analytical modeling of *all* unmeasured confounding, closed-form derivation of the bias function (Lemma 4.2). | **Yes.** Our method works with non-binary/continuous $U, S$, as we show in our experiments in Appendix B.1. Multivariate $U, S$ also does not constitute any problem. |
| Distribution of $U$: $P(U = 1|R=r)=1/2, r\in\{0,1\}$ | Simple bias expressions, assumed for analytical convenience. | **Yes.** Does not affect practice. Theoretical results can also be extended to different $U$ distributions. |
| Binary treatment $A$ and outcome $Y$ variables | Common assumptions in literature. Primarily made for analytical tractability. | **Yes.** As long as a measure of "predictive error" in nuisance estimators can be calculated, practical implementation remains feasible. |
| Exogeneity: $U \perp X$ (Assumption 4.1) | Isolates the effect of $U$ from measured covariates $X$. | **Yes.** The method would capture the signal from the part of $U$ not explained by $X$. |
| Weak transportability and ignorability (Assumption 4.1) | Derivation of the bias function analytically (Lemma 4.2), and study how it changes with $X$. | **Yes.** Since $U$ is not assumed to be observed and is not a part of our estimators, these assumptions have limited effect in practice. |
| Data-generating process (Algorithm 1 & Eq. 6-7) | To encode low-uncertainty in target variables. Key to results in Table 1. Supported with real data (Figures 2, 13). | **Partially.** Our most substantive assumption. As long as a low-uncertainty distribution (on $S, A, Y$, given $X$) can be assumed for $\mathcal{F}(p)$, the insights from Table 1 still apply. |
| Single source of bias (including Assumption 4.5) | We guarantee identification when only one of the four bias mechanisms in Figure 1 is at play. | **Partially.** Under multiple biases, interpretation of signals changes, but remains useful. Section B.1 illustrates synthetic and real-world examples. |

picture. Specifically, they miss *contextual* independencies. Consider Figure 1b where $U$ influences the treatment $A$ *in general*. However, we might have

$$A \perp\!\!\!\perp U \mid X = x_m, R = 0, \quad \text{whereas}$$
$$A \not\!\!\perp\!\!\!\perp U \mid X = x_n, R = 0. \tag{8}$$

That is, $U$ affects $A$ only for *some* patients, $X = x_m$, but not others, $X = x_n$ (as in our clinical vignette). We discuss a direct result of this observation next.

**The bias in the OS varies across patients.** Contextual independence of $U$ and $T \in \{S, A, Y^0, Y^1\}$ leads to a key observation: the magnitude of bias will vary across patients. Consider Figure 1b and assume that $U$ has an effect on $Y^1$ for *all* $X = x$. That is, $Y^1 \not\!\!\perp\!\!\!\perp U \mid X, R = 0$. We have, together with Eq. (8),

$$Y^1 \perp\!\!\!\perp A \mid X = x_m, R = 0, \quad \text{whereas}$$
$$Y^1 \not\!\!\perp\!\!\!\perp A \mid X = x_n, R = 0. \tag{9}$$

since the "ignorability of treatment" condition in Assumption 2.1 holds for $X = x_m$, but not for $X = x_n$ in the OS ($R = 0$). One then has, $f_1(x_m) = \text{CATE}_{\text{os}}(x_m, A = 1)$ but $f_1(x_n) \neq \text{CATE}_{\text{os}}(x_n, A = 1)$. Consequently, when transportability bias is not an issue, we have $b_1(x_m) = 0$, but $b_1(x_n) \neq 0$.

More generally, for patients where the effect of $U$ on $T \in \{S, A, Y^0, Y^1\}$ is stronger, we will observe two key phenomena: (i) the magnitude of the bias, $|b(X)|$, shall be larger. (ii) Large-bias regions also exhibit greater variability in $T$, due to the residual influence of $U$.

*Table 2.* Covariance between the magnitude of bias and conditional variances of downstream variables under different bias mechanisms (see Theorems 4.4 and 4.6).

| Bias Type | $\bar{\rho}(b_1, S)$ | $\bar{\rho}(b_1, A)$ | $\bar{\rho}(b_1, Y)$ |
|---|---|---|---|
| No Bias | $= 0$ | $= 0$ | $= 0$ |
| Transportability | $= 0$ | $= 0$ | $> 0$ |
| Confounding | $= 0$ | $> 0$ | $> 0$ |
| Selection Type 1 | $> 0$ | $= 0$ | $> 0$ |
| Selection Type 2 | $\neq 0$ | $\neq 0$ | $\neq 0$ |

For instance, with confounding bias in Figure 1b, we expect *larger variances* in $A$ and $Y$ in *severely biased regions*. One can formalize this idea through the *covariances* of the absolute bias function and conditional variance functions: $\mathrm{Cov}\big(|b_1(X)|, \mathrm{Var}(A|X, S = 1)\big)$ and $\mathrm{Cov}\big(|b_1(X)|, \mathrm{Var}(Y|X, S{=}1, A{=}1)\big)$.

### 4.3. Covariance of the Bias and Conditional Variances in Downstream Variables

We analyze the covariance between the magnitude of bias, $|b_a(X)|$, and the conditional variances of $S$, $A$, and $Y$, under different bias mechanisms in Figure 1. Our main result is that covariance signals construct a "hash table", allowing us to differentiate between the bias mechanisms (Table 2).

Table 2 is derived assuming only one bias mechanism is present. When multiple biases are present, the covariance signal patterns need not identify a unique source of bias. In such cases, we interpret the signals as evidence about plausible dominant mechanisms rather than as an exhaustive classifier. We study multiple-bias settings synthetically in Section B.1 and in a real-world study in Section 6.

For simplicity, we focus on treatment $A = 1$ and define

$$p_u^T := P(T{=}1|X, U{=}u, R{=}0),\ T \in \{S, A, Y^1\}. \quad (10)$$

$$p_r^U := P(U{=}1|R{=}r). \quad (11)$$

**Lemma 4.2** (Quantifying the Bias). *Suppose that Assumptions 2.1 and 4.1 hold, and that $p_r^U = 1/2$ for $r \in \{0, 1\}$ (see Eq. (11)).*

1. **Transportability Bias**– *Consider Figure 1a where $S, A$, and $U$ are independent of each other given $X$, in the OS ($R = 0$). We have,*

$$b_1(X) = (p_{r=1}^U - p_{r=0}^U)(p_{u=1}^Y - p_{u=0}^Y). \quad (12)$$

2. **Confounding Bias**– *Consider Figure 1b where $S \perp\!\!\!\perp A, U \mid X, R = 0$ and assume that $p_{r=1}^U = p_{r=0}^U = 1/2$.*

*We have,*

$$b_1(X) = \frac{(p_{u=1}^Y - p_{u=0}^Y)(p_{u=1}^A - p_{u=0}^A)}{2(p_{u=1}^A + p_{u=0}^A)}. \quad (13)$$

3. **Selection Bias, Type 1**– *Consider Figure 1c where $A \perp\!\!\!\perp S, U \mid X, R = 0$ and assume that $p_{r=1}^U = p_{r=0}^U = 1/2$. We have,*

$$b_1(X) = \frac{(p_{u=1}^Y - p_{u=0}^Y)(p_{u=1}^S - p_{u=0}^S)}{2(p_{u=1}^S + p_{u=0}^S)}. \quad (14)$$

The results are intuitive. In Eq. (12), bias is larger if the distribution of $U$ in the RCT and OS differ significantly, *i.e.,* $p_{r=1}^U - p_{r=0}^U$ is big (weak transportability). The contrast between the outcomes for different values of $U$, *i.e.,* $p_{u=1}^Y - p_{u=0}^Y$, is another driver of the bias. Confounding and selection biases react similarly to the strength of interaction between $U$ and $S, A, Y$.

Building on Lemma 4.2, we derive covariance relations between $b_1(X)$ and the conditional variances of downstream variables $S, A, Y$. Let us first introduce a shorthand notation for the covariance signals.

**Definition 4.3** (Covariance signals).

$$\bar{\rho}(b_1, S) := \mathrm{Cov}\big(|b_1(X)|, \mathrm{Var}(S|X, R{=}0)\big).$$
$$\bar{\rho}(b_1, A) := \mathrm{Cov}\big(|b_1(X)|, \mathrm{Var}(A|X, S{=}1, R{=}0)\big).$$
$$\bar{\rho}(b_1, Y) := \mathrm{Cov}\big(|b_1(X)|, \mathrm{Var}(Y|X, S{=}1, A{=}1, R{=}0)\big).$$

**Theorem 4.4.** *Suppose that Assumptions 2.1 and 4.1 hold, and downstream variables $T \in \{S, A, Y^0, Y^1\}$ are sampled according to Algorithm 1 sfor some $\mathcal{F}(p) \in \mathcal{F}$. For confounding and type-1 selection biases (Figures 1b, 1c), assume that $p_r^U = 1/2$ (see Eq. (11)) for simplicity. For transportability bias (Figure 1a), let $p_{r=0}^U \neq p_{r=1}^U$ and $p_r^U \sim \mathcal{F}(p)$. Then, these three bias mechanisms are uniquely characterized by the covariance signals (see Definition 4.3) as in Table 2.*

### 4.4. Covariances Under Type 2 Selection Bias

The bias mechanisms in Figures 1a-1c originate from an *unmeasured* covariate $U$. This commonality enabled us to employ consistent analytical approaches to derive the bias expressions (Lemma 4.2) and covariance signals (Theorem 4.4). Here, we investigate type 2 selection bias (Figure 1d), where the root cause is not an unmeasured covariate, but the selection variable $S$ acting as a collider instead.

This form of bias, while challenging to detect, is prevalent. Understanding how covariances behave under its various manifestations is therefore crucial. First, given the distinct graph structure (absence of $U$), we present a modified version of Assumption 4.1.

**Assumption 4.5.** We have *transportability* $Y^a \perp\!\!\!\perp R|X$; and *ignorability* $Y^a \perp\!\!\!\perp A|X, R$ for $a \in \{0,1\}$.

Note that due to the $A \rightarrow S$ and $Y \rightarrow S$ edges, ignorability does not hold: $Y^a \not\perp\!\!\!\perp A \mid X, R, S$, which leads to bias as $f_a(X) \neq \text{CATE}_{\text{os}}(x,a)$ (see Eq. (5)).

Covariance signals in this case depend on the specific selection probabilities: $P(S = 1|X, Y, A)$. We defer the bulk of the technical analysis to Section A.3 and give the main result directly, which states that the covariance signals will be non-zero in general.

**Theorem 4.6.** *Suppose that Assumption 4.5 holds and the downstream variables $T \in \{S, A, Y^0, Y^1\}$ are sampled according to Algorithm 1 for some distribution $\mathcal{F}(p) \in \mathcal{F}$. Consider type 2 selection bias in Figure 1d where $Y^a \not\perp\!\!\!\perp A \mid X, S, R = 0$. Then, covariance signals (see Definition 4.3) will be non-zero in general.*

For example, consider $P(S = 1|Y = 1, A = 1) = 0.9$ and $P(S = 1|Y = 0, A = 1) = 0.1$, where there is preferential selection of patients who experience the outcome. We discuss why this yields a positive signal $\bar{\rho}(b_1, Y)$. This selection mechanism systematically overestimates event rates. Importantly, bias magnitude will vary by baseline risk: For those with low event rates, *e.g.*, $P(Y^1 = 1|X = x_1) = 0.1$, the estimate will gravitate toward $0.5$ (potentially large bias) where the variance is higher. Conversely, in high-risk groups, *e.g.*, $P(Y^1 = 1|X = x_2) = 0.9$, the bias is smaller (at most $0.1$) and the variance in outcomes will also be smaller.

### 4.5. Consistent Estimators of Covariance

Estimating the covariances is not straightforward, particularly when $X$ contains continuous variables. We construct an estimator based on instance-wise squared-errors, which is strongly consistent when the nuisance function estimators in the OS are strongly consistent.

**Definition 4.7** (Nuisance estimators in the OS)**.**

$$\widehat{\eta}_S(X) := \widehat{P}(S = 1 \mid X, R = 0). \tag{15}$$

$$\widehat{\eta}_A(X) := \widehat{P}(A = 1 \mid X, S = 1, R = 0). \tag{16}$$

$$\widehat{\eta}_Y(X) := \widehat{P}(Y = 1 \mid X, S = 1, A = 1, R = 0). \tag{17}$$

$$\widehat{b}_1(X) := \widehat{g}_1(X) - \widehat{f}_1(X), \tag{18}$$

where $g_1(X)$ and $f_1(X)$ are defined in Eq. (3) and (4). $\widehat{g}_1(X)$ is estimated from the RCT, and the remaining are estimated from the OS. We use the following estimator, which is for the covariance between the bias function and the *squared-error* of the nuisance function estimators for

the target variables $T \in \{S, A, Y\}$.

$$\widehat{\bar{\rho}(b_1, T)} = \frac{n}{n-1}\left(\frac{1}{n}\sum_{i=1}^{n}|\widehat{b}_1(X_i)|(T_i - \widehat{\eta}_T(X_i))^2 \right.$$
$$\left. - \frac{1}{n^2}\sum_{i=1}^{n}\sum_{j=1}^{n}|\widehat{b}_1(X_i)|(T_j - \widehat{\eta}_T(X_i))^2\right).$$

**Theorem 4.8.** *Assume that the estimators in Eq. (15)-Eq. (18) are strongly consistent, and are dominated by an integrable function $H(X)$. Then, for all $T \in \{S, A, Y\}$, $\widehat{\bar{\rho}(b_1, T)}$ are asymptotically unbiased and strongly consistent estimators of $\bar{\rho}(b_1, T)$.*

## 5. Synthetic Experiments

The synthetic experiments serve two purposes. First, they verify that the empirical covariance estimates recover the population-level signatures predicted by Table 2 under the generative model used in the analysis. Second, the additional experiments in Section B.1 stress-test the diagnostic signals beyond the simplest theoretical setting, including smaller RCT sample sizes, different dimensions of $X$, continuous unmeasured covariates, alternative type-2 selection mechanisms, and mixtures of multiple bias types. These experiments do not provide formal guarantees outside the stated assumptions, but they help characterize when the proposed signals remain informative in more realistic settings.

We run several synthetic experiments under the generative model in Section 4. We consider binary covariates, $X \in \{0,1\}^d$, and denote by $X(j)$ the $j$-th covariate. We experiment with $d \in \{5, 6, 7\}$. We set $P(X(j) = 1|R = 1) = 0.4$ in RCT and $P(X(j) = 1|R = 0) = 0.6$ in OS to simulate covariate shift between two populations. In the RCT cohort, we use $P(A = 1|R = 1) = 0.5$ and $P(S = 1|R = 1) = 1$. We set the OS cohort size to $n_{\text{os}} = 50000$ for fitting $\hat{f}_1(X)$, and set aside $n_{\text{val}} = 2000$ OS participants to estimate the correlation signals in Definition 4.3 once the nuisance functions in Eq.15 - 18 are fitted. We experiment with different RCT cohort sizes, $n_{\text{rct}} \in \{2000, 50000\}$, for fitting $\hat{g}_1(X)$.

For each bias mechanism in Figure 1, we make 200 independent runs, where we sample a $p \sim \text{Uniform}[0.2, 0.5]$ and specify $\mathcal{F}(p)$ in Algorithm 1 accordingly, from which downstream variables are sampled.

When simulating transportability bias in Figure 1a, we sample $P(U = 1 \mid X, R) \sim \mathcal{F}(p)$ to induce a different distribution of $U$ in the RCT and OS. We set $U$-bias to `False` for $S, A$ and `True` for $Y^1$ in Algorithm 1.

In other experiments, we set $P(U \mid X, R) = 1/2$ to ensure transportability. When simulating confounding bias in Figure 1b, we set $U$-bias to `False` for $S$ and `True` for $A, Y^1$

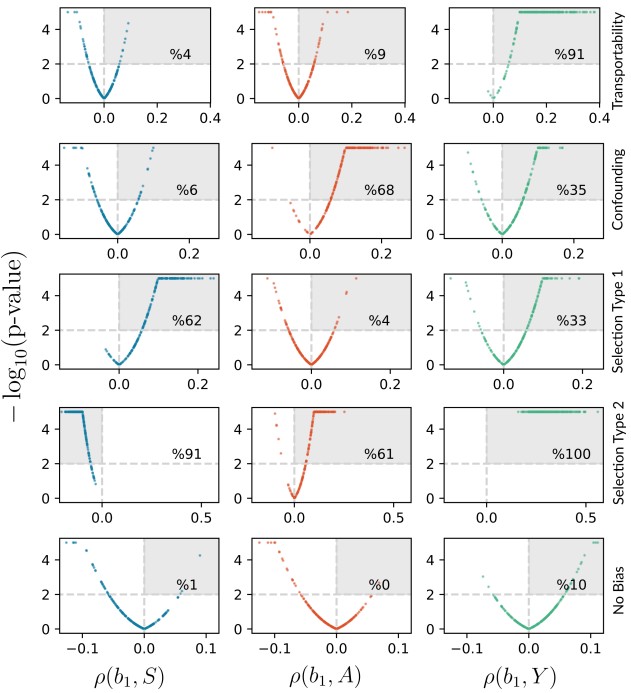

*Figure 3.* Pearson's R (normalized covariance) values in synthetic experiments. Percentages denote the ratio runs in shaded area. $p$-values clipped at $p = 10^{-5}$.

in Algorithm 1. For type 1 selection bias in Figure 1c, we set $U$-bias to False for $A$ and True for $S, Y^1$. Finally, for type 2 selection bias in Figure 1d, we set $U$-bias to False for $A$ and $Y^1$. We set $P(S=1|Y=1, A=1) = 0.9$ and to 0.1 for other combinations of $Y$ and $A$. In Section A.3, we study different parametrizations of type 2 selection bias.

We set $n_{\text{rct}} = 50000$ and $d = 6$. The results for $n_{\text{rct}} = 2000$ and $d \in \{5, 7\}$ are presented in Section B.1. The (Pearson's R, $p$-value) pairs computed from 200 experiment runs are presented in Figure 3 (each dot is a run). We draw a nominal threshold for the significance of the estimated correlation coefficients at $p = 0.01$.

The percentages shown in Figure 3 are computed across the 200 independent simulation runs for each bias setting. They report the fraction of runs in which the estimated correlation falls in the theoretically predicted region and is statistically significant at the nominal threshold $p < 0.01$. These percentages should not be interpreted as effect sizes or as thresholds for deciding whether a single observational study contains bias. Rather, they summarize the finite-sample alignment between the empirical covariance signals and the qualitative patterns predicted by Table 2. Thus, a smaller percentage for a signal predicted to be nonzero indicates that this particular covariance is harder to detect reliably in that setting, not that the reported percentage itself is a measure of statistical significance.

The results corroborate our findings: correlations between magnitude of the bias, $|b_1(X)|$, and squared errors of the nuisance function estimators (see Definition 4.7) can characterize the underlying bias mechanism as laid out in Table 2.

We experiment with *multiple* bias mechanisms in Section B.1. Further, we report the results for experiments where $U$ is continuous, which yield similar observations, showing that empirical insights from our methods generalize beyond the settings for which we derive our theory.

Finally, the reliability of these empirical signatures depends on the amount of information available to estimate both the bias function and the nuisance functions. In additional experiments in Section B.1, we find that the covariance signals become weaker as the RCT sample size decreases or as the dimension of $X$ increases. This is expected: estimating $\hat{g}_1(X)$ from the RCT and the nuisance functions from the OS becomes more difficult in smaller or higher-dimensional settings, which reduces the power to detect the predicted covariance patterns. Thus, while Table 2 describes population-level diagnostic signatures, finite-sample use of the method should account for uncertainty in the estimated signals, especially when RCT sample sizes are limited.

## 6. Real-world Experiments

The OS and RCT components of Women's Health Initiative (WHI) yielded conflicting results regarding the effect of combined hormonal therapy (HRT) on coronary heart disease (CHD) and stroke outcomes, where the OS suggested a protective effect while the RCT revealed increased risk. Follow-up studies found that the might have OS suffered from "immortal time" bias, which can broadly be thought of as a subcategory of type 2 selection bias (Prentice et al., 2005) (see Section 3.2).

**Setup** Following Prentice et al. (2005), we attempt to correct for the immortal time bias by *only* selecting patients into the "treatment group" who were not past users of combined HRT but started shortly after enrollment (see Section B.2 for our reproduction of results). We refer to the setting prior to correction as the "Baseline" analysis. We refer to the setting after the correction as "Corrected," although it may not be fully unbiased as other types of biases may still be present. Finally, we run experiments where we conceal age and menopausal status in the "Corrected" setting and refer to this setting as "Manual bias".

To compute the correlation signals, we have the following definitions of $S$ and $Y$: $Y = 1$ if the patient experiences the event within the follow-up period, and $Y = 0$ otherwise. $S = 0$ for patients who are censored or past HRT users who do not meet the inclusion criteria, and $S = 1$ for those included in the analysis.

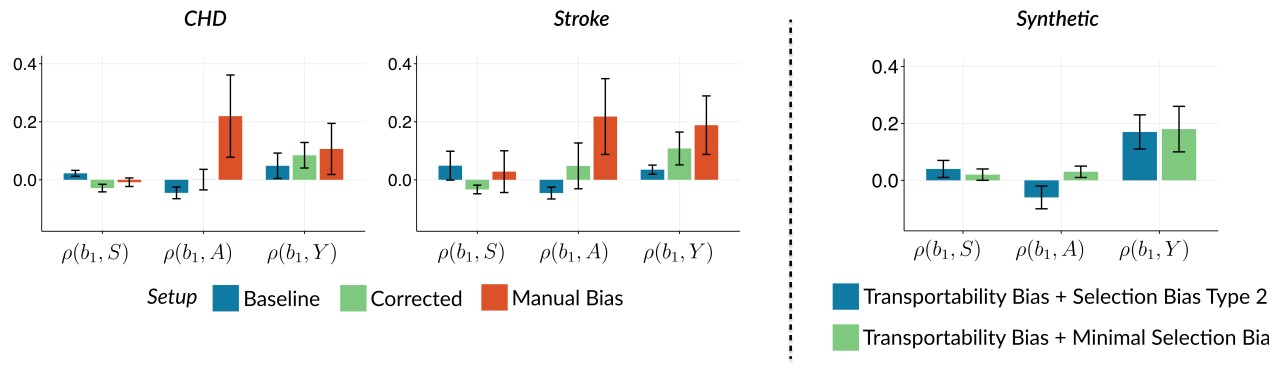

*Figure 4.* **Left**: Pearson's R between bias functions and downstream variables for CHD and stroke in WHI experiments. **Right**: Average correlation signals under different synthetic settings: 1) type 2 selection bias (with selection probabilities matching WHI setting) and transportability bias, 2) minimal selection bias.

**Results** In Figure 4 (left), we show covariance signals for all cases. 95% confidence intervals are computed using 20 train/validation splits of the RCT and OS.

In baseline analyses, we see a type 2 selection pattern: $\rho(b_1, S) > 0$, $\rho(b_1, A) < 0$, $\rho(b_1, Y) > 0$. Focusing on $\rho(b_1, A)$, we note the treated group in the OS is largely composed of patients more likely to benefit from the treatment, which explains the high bias and low variance in their management, hence a negative $\rho(b_1, A)$.

As we implement the bias-correction step, the $\rho(b_1, S)$ and $\rho(b_1, A)$ signals tend to deflate, as expected. A more subtle observation is that $\rho(b_1, Y)$ does not decrease. We hypothesize that this may reflect the presence of two bias mechanisms: type-2 selection bias and transportability bias. If the correction primarily removes the selection component, then the remaining bias may be driven more by transportability. One may then observe an increase, or no decrease, in $\rho(b_1, Y)$ after correction, since transportability bias is expected to mostly align with outcome-prediction uncertainty.

To probe this hypothesis, we conduct a synthetic experiment that combines type-2 selection bias with transportability bias, using selection probabilities chosen to mimic the WHI setting ($p_{00}^S = 0.9$, $p_{01}^S = 0.9$, $p_{10}^S = 0.3$, $p_{11}^S = 0.1$), as shown in Figure 4 (right). The combined synthetic setting reproduces the qualitative signal pattern observed in the real-world "Baseline" WHI analysis. Moreover, after correcting only for selection bias, $\rho(b_1, A)$ and $\rho(b_1, S)$ move toward zero, while $\rho(b_1, Y)$ slightly increases, mirroring the behavior seen in the real-world results. Further details of this experiment are provided in Section B.2.

Finally, we conduct *positive-control* experiments, manually concealing age and menopausal status, which are confounders for treatment assignment and outcome ("Manual Bias" in Figure 4). As expected, we note a significant boost in $\rho(b_1, A) > 0$ and $\rho(b_1, Y) > 0$.

## 7. Conclusion and Limitations

We uncover a novel connection between the predictive performance of nuisance function estimators and bias in observational studies, enabling identification of the underlying bias mechanism. More broadly, our results suggest that model performance diagnostics can be leveraged to diagnose flaws in observational study design. Our analyses rely on a specific generative model, which is a limitation. However, we empirically support our data-generating assumptions in real-world data (see Table 1) and show that the resulting insights extend beyond this theoretical setting (see Section B.1). Another limitation is the dependence on patient-level RCT data. Removing this requirement is a nontrivial future work. Finally, we do not address measurement bias—systematic errors in variable observation—which remains an important avenue for future study.

Our covariance-based statistic is not the only possible measure of the relation between bias and nuisance prediction error. We use it because it is simple and analytically tractable, but other nonlinear measures could be considered. Similarly, while our plug-in covariance estimators are consistent under consistent nuisance estimation, we do not establish semiparametric efficiency or optimality guarantees. Developing more efficient, orthogonal, or cross-fitted estimators of these signals is an important direction for future work.

## Acknowledgments

ID and DS were supported by Office of Naval Research Award No. N00014-21-1-2807. ZH was supported by the National Cancer Institute of the National Institutes of Health under Award Number F30CA268631. The content is solely the responsibility of the authors and does not necessarily represent the official views of the National Institutes of Health. PDB was supported by the Hasler Foundation grant number 2105.

## Impact Statement

This paper advances methodological understanding in causal inference by proposing a principled framework for diagnosing bias mechanisms in observational studies using model performance diagnostics. By improving researchers' ability to distinguish among sources of bias such as confounding, selection bias, and lack of transportability, our work has the potential to improve the reliability and interpretability of observational analyses, particularly in high-stakes domains such as healthcare. While the methods are intended to support better study design and analysis rather than automated decision-making, incorrect use or over-interpretation could still lead to misleading conclusions if underlying assumptions are violated. We therefore emphasize the importance of careful validation and domain expertise when applying our approach. Overall, we believe the societal implications of this work are aligned with well-established goals in machine learning and causal inference, namely improving the robustness and credibility of data-driven scientific findings.

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

# A. Proofs

Detailed proofs for the theoretical results in the main paper are provided here. All results in the main paper are re-stated in this section for clarity.

## A.1. Useful Lemmas

We start by deriving some auxiliary results that hold true under the general weak Assumption 4.1 we make throughout the paper.

**Lemma A.1.** *Under Assumption 4.1, we have*

$$
f_a(X)
$$
$$
= \sum_{u=0,1} P(Y = 1, U = u \mid X, R = 0, S = 1, A = a)
$$
$$
= \sum_{u=0,1} P(Y^a = 1 \mid X, U = u, R = 0, S = 1, A = a) P(U = u \mid X, R = 0, S = 1, A = a)
$$
$$
= \sum_{u=0,1} P(Y^a = 1 \mid X, U = u, R = 0) P(U = u \mid X, R = 0, S = 1, A = a) \tag{19}
$$
$$
= \sum_{u=0,1} P(Y^a = 1 \mid X, U = u) \frac{P(U = u, S = 1, A = a \mid X, R = 0)}{P(S = 1, A = a \mid X, R = 0)} \tag{20}
$$
$$
= \sum_{u=0,1} P(Y^a = 1 \mid X, U = u) \frac{P(S = 1, A = a \mid X, U = u, R = 0) P(U = u \mid R = 0)}{P(S = 1, A = a \mid X, R = 0)} \tag{21}
$$
$$
= \sum_{u=0,1} P(U = u \mid R = 0) P(Y^a = 1 \mid X, U = u)
$$
$$
\times \frac{P(S = 1 \mid X, U = u, R = 0) P(A = a \mid X, U = u, R = 0)}{P(S = 1, A = a \mid X, R = 0)}. \tag{22}
$$

*Eq. (19) follows from weak ignorability, Eq. (20) from weak transportability, Eq. (21) from exogeneity of $X$ and $U$, and Eq. (22) from weak ignorability again.*

**Lemma A.2.** *Under Assumption 4.1, we have*

$$
P(S = 1 \mid X, R = 0)
$$
$$
= P(S = 1, U = 0 \mid X, R = 0) + P(S = 1, U = 1 \mid X, R = 0)
$$
$$
= P(S = 1 \mid X, U = 0, R = 0) P(U = 0 \mid R = 0)
$$
$$
+ P(S = 1 \mid X, U = 1, R = 0) P(U = 1 \mid R = 0) \tag{23}
$$
$$
= p_{u=0}^{S}(1 - p_{r=0}^{U}) + p_{u=1}^{S} p_{r=0}^{U}
$$
$$
= p_{u=0}^{S} + p_{r=0}^{U}(p_{u=1}^{S} - p_{u=0}^{S}) \tag{24}
$$

*where Eq. (23) follows from exogeneity of $X$ and $U$, and Eq. (24) from Eq. (10), Eq. (11).*

**Lemma A.3.** *Under Assumption 4.1, we have*

$$
P(A = 1 \mid X, S = 1, R = 0)
$$
$$
= P(A = 1, U = 0 \mid X, S = 1, R = 0) + P(A = 1, U = 1 \mid X, S = 1, R = 0)
$$
$$
= P(A = 1 \mid X, U = 0, S = 1, R = 0) P(U = 0 \mid X, S = 1, R = 0)
$$
$$
+ P(A = 1 \mid X, U = 1, S = 1, R = 0) P(U = 1 \mid X, S = 1, R = 0)
$$
$$
= P(A = 1 \mid X, U = 0, R = 0) P(U = 0 \mid X, S = 1, R = 0)
$$
$$
+ P(A = 1 \mid X, U = 1, R = 0) P(U = 1 \mid X, S = 1, R = 0) \tag{25}
$$
$$
= p_{u=0}^{A} P(U = 0 \mid X, S = 1, R = 0) + p_{u=1}^{A} P(U = 1 \mid X, S = 1, R = 0) \tag{26}
$$

*where Eq. (25) follows from weak ignorability, and Eq. (26) from Eq. (10).*

**Lemma A.4.** *Under Assumption 4.1, we have*

$$
\begin{aligned}
& P(Y = 1 \mid X, S = 1, A = 1, R = 0) \\
&= P(Y = 1, U = 0 \mid X, S = 1, A = 1, R = 0) + P(Y = 1, U = 1 \mid X, S = 1, A = 1, R = 0) \\
&= P(Y^1 = 1 \mid X, U = 0, S = 1, A = 1, R = 0)P(U = 0 \mid X, S = 1, A = 1, R = 0) \\
&\quad + P(Y^1 = 1 \mid X, U = 1, S = 1, A = 1, R = 0)P(U = 1 \mid X, S = 1, A = 1, R = 0) \\
&= p_{u=0}^Y P(U = 0 \mid X, S = 1, A = 1, R = 0) + p_{u=1}^Y P(U = 1 \mid X, S = 1, A = 1, R = 0)
\end{aligned}
\tag{27}
$$

*where Eq. (27) follows from weak ignorability.*

## A.2. Proofs of the Results in Section 4.3

**Lemma A.5** (Quantifying the Bias)**.** *Suppose that Assumptions 2.1 and 4.1 hold, and that $p_r^U = 1/2$ for $r \in \{0, 1\}$ (see Eq. (11)).*

1. ***Transportability Bias**– Consider Figure 1a where $S$, $A$, and $U$ are independent of each other given $X$, in the OS ($R = 0$). We have,*

$$
b_1(X) = (p_{r=1}^U - p_{r=0}^U)(p_{u=1}^Y - p_{u=0}^Y).
\tag{12}
$$

2. ***Confounding Bias**– Consider Figure 1b where $S \perp\!\!\!\perp A, U \mid X, R = 0$ and assume that $p_{r=1}^U = p_{r=0}^U = 1/2$. We have,*

$$
b_1(X) = \frac{(p_{u=1}^Y - p_{u=0}^Y)(p_{u=1}^A - p_{u=0}^A)}{2(p_{u=1}^A + p_{u=0}^A)}.
\tag{13}
$$

3. ***Selection Bias, Type 1**– Consider Figure 1c where $A \perp\!\!\!\perp S, U \mid X, R = 0$ and assume that $p_{r=1}^U = p_{r=0}^U = 1/2$. We have,*

$$
b_1(X) = \frac{(p_{u=1}^Y - p_{u=0}^Y)(p_{u=1}^S - p_{u=0}^S)}{2(p_{u=1}^S + p_{u=0}^S)}.
\tag{14}
$$

*Proof of the Transportability Bias.* We consider Figure 1a where $S \perp\!\!\!\perp A \perp\!\!\!\perp U \mid X, R = 0$. That is, $X$ explains everything in the OS that we are interested in. However, we allow $U \not\perp\!\!\!\perp R$, that is, the unmeasured covariate has different distributions in the RCT and OS. Following Eq. (22), we have

$$
f_a(X) = \sum_{u=0,1} P(U = u \mid R = 0)P(Y^a = 1 \mid X, U = u)
\tag{28}
$$

$$
\times \frac{P(S = 1 \mid X, U = u, R = 0)P(A = a \mid X, U = u, R = 0)}{P(S = 1 \mid X, R = 0)P(A = a \mid X, R = 0)} \quad (S \perp\!\!\!\perp A \perp\!\!\!\perp U \mid X, R = 0)
$$

$$
= \sum_{u=0,1} P(Y^a = 1 \mid X, U = u)P(U = u \mid R = 0).
\tag{29}
$$

Combining Eq. (33) and Eq. (29) we have

$$
\begin{aligned}
b_a(X) &= g_a(X) - f_a(X) \\
&= P(Y^a = 1 \mid X, U = 0)\big(P(U = 0 \mid R = 1) - P(U = 0 \mid R = 0)\big) \\
&\quad + P(Y^a = 1 \mid X, U = 1)\big(P(U = 1 \mid R = 1) - P(U = 1 \mid R = 0)\big) \\
&= \big(P(Y^a = 1 \mid X, U = 1) - P(Y^a = 1 \mid X, U = 0)\big) \\
&\quad \big(P(U = 1 \mid R = 1) - P(U = 1 \mid R = 0)\big) \\
&= \big(p_{r=1}^U - p_{r=0}^U\big)\big(p_{u=1}^Y - p_{u=0}^Y\big).
\end{aligned}
\tag{30}
$$

The result then follows from Eq. (36) and Eq. (10),Eq. (11) for $a = 1$. $\qquad \square$

*Proof of the Confounding Bias.* We consider Figure 1b where $S \perp\!\!\!\perp A, U \mid X, R = 0$ and $P(U = 0 \mid R = 0) = P(U = 1 \mid R = 0) = 1/2$. Following Eq. (22), we have

$$
\begin{aligned}
f_a&(X) \\
&= \frac{1}{2} \sum_{u=0,1} P(Y^a = 1 \mid X, U = u) \frac{P(S = 1 \mid X, U = u, R = 0) P(A = a \mid X, U = u, R = 0)}{P(S = 1 \mid X, R = 0) P(A = a \mid X, R = 0)} \quad (S \perp\!\!\!\perp A \mid X, R = 0) \\
&= \frac{1}{2} \sum_{u=0,1} P(Y^a = 1 \mid X, U = u) \frac{P(A = a \mid X, U = u, R = 0)}{P(A = a \mid X, R = 0)} \quad (S \perp\!\!\!\perp U \mid X, R = 0) \\
&= \frac{1}{2} \sum_{u=0,1} P(Y^a = 1 \mid X, U = u) \frac{P(A = a \mid X, U = u, R = 0)}{\sum_{u=0,1} P(A = a, U = u \mid X, R = 0)} \\
&= \frac{1}{2} \sum_{u=0,1} P(Y^a = 1 \mid X, U = u) \frac{P(A = a \mid X, U = u, R = 0)}{\sum_{u=0,1} P(A = a \mid X, U = u, R = 0) P(U = u \mid X, R = 0)} \\
&= \frac{\cancel{1}}{\cancel{2}} \sum_{u=0,1} P(Y^a = 1 \mid X, U = u) \frac{P(A = a \mid X, U = u, R = 0)}{\sum_{u=0,1} P(A = a \mid X, U = u, R = 0) \underbrace{P(U = u \mid R = 0)}_{\cancel{1/2}}} \\
&= \frac{\Big(P(Y^a = 1 \mid X, U = 0) P(A = a \mid X, U = 0, R = 0) + P(Y^a = 1 \mid X, U = 1) P(A = a \mid X, U = 1, R = 0)\Big)}{P(A = a \mid X, U = 0, R = 0) + P(A = a \mid X, U = 1, R = 0)}.
\end{aligned}
\tag{31}
$$

Next, note that

$$
\begin{aligned}
g_a(X) &= P(Y^a = 1 \mid X, R = 1) \\
&= \sum_{u=0,1} P(Y^a = 1, U = u \mid X, R = 1) \\
&= \sum_{u=0,1} P(Y^a = 1 \mid X, U = u, R = 1) P(U = u \mid X, R = 1) \\
&= \sum_{u=0,1} P(Y^a = 1 \mid X, U = u, R = 1) P(U = u \mid R = 1) \tag{32} \\
&= \sum_{u=0,1} P(Y^a = 1 \mid X, U = u) P(U = u \mid R = 1) \tag{33} \\
&= \frac{P(Y^a = 1 \mid X, U = 0) + P(Y^a = 1 \mid X, U = 1)}{2}. \tag{34}
\end{aligned}
$$

where Eq. (32) and Eq. (33) follow from the exogeneity of $X$ and $U$ and weak transportability, respectively, in Assumption 4.1. Eq. (34) follows from the distribution of $U$ given above.

Combining Eq. (31) and Eq. (34), we have

$$
\begin{aligned}
b_a(X) &= g_a(X) - f_a(X) \\
&= \frac{\big(P(Y^a = 1 \mid X, U = 0) - P(Y^a = 1 \mid X, U = 1)\big) \times \big(P(A = a \mid X, U = 1, R = 0) - P(A = a \mid X, U = 0, R = 0)\big)}{2\big(P(A = a \mid X, U = 1, R = 0) + P(A = a \mid X, U = 0, R = 0)\big)}
\end{aligned}
\tag{35}
$$

The result then follows from Eq. (35) and Eq. (10) for $a = 1$. $\qquad\square$

*Proof of the Selection Bias.* We consider Figure 1b where $A \perp\!\!\!\perp S, U \mid X, R = 0$ and $P(U = 0 \mid R = 0) = P(U = 1 \mid$

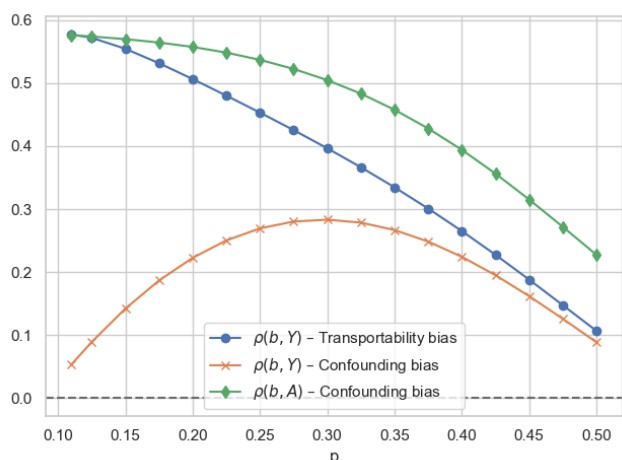

*Figure 5.* Different values of the correlation between the absolute bias function and the variance of downstream variables for different values of $p \in (0.1, 0.5)$ that parametrize $\mathcal{F}(p)$ in Eq. (6).

$R = 0) = 1/2$. Following Eq. (22), we have

$$
\begin{aligned}
& f_a(X) \\
&= \frac{1}{2} \sum_{u=0,1} P(Y^a = 1 \mid X, U = u) \frac{P(S = 1 \mid X, U = u, R = 0) P(A = a \mid X, U = u, R = 0)}{P(S = 1 \mid X, R = 0) P(A = a \mid X, R = 0)} && (A \perp\!\!\!\perp S \mid X, R = 0) \\
&= \frac{1}{2} \sum_{u=0,1} P(Y^a = 1 \mid X, U = u) \frac{P(S = 1 \mid X, U = u, R = 0)}{P(S = 1 \mid X, R = 0)}. && (A \perp\!\!\!\perp U \mid X, R = 0)
\end{aligned}
$$

Rest of the steps follow similarly to confounding bias case and we have

$$
\begin{aligned}
b_a(X) &= g_a(X) - f_a(X) \\
&= \frac{\begin{aligned}&\left(P(Y^a = 1 \mid X, U = 0) - P(Y^a = 1 \mid X, U = 1)\right) \\ &\quad \times \left(P(S = 1 \mid X, U = 1, R = 0) - P(S = 1 \mid X, U = 0, R = 0)\right)\end{aligned}}{2\left(P(S = 1 \mid X, U = 1, R = 0) + P(S = 1 \mid X, U = 0, R = 0)\right)}.
\end{aligned} \tag{36}
$$

The result then follows from Eq. (36) and Eq. (10) for $a = 1$. $\qquad\square$

**Theorem 4.4.** *Suppose that Assumptions 2.1 and 4.1 hold, and downstream variables $T \in \{S, A, Y^0, Y^1\}$ are sampled according to Algorithm 1 sfor some $\mathcal{F}(p) \in \mathcal{F}$. For confounding and type-1 selection biases (Figures 1b, 1c), assume that $p_r^U = 1/2$ (see Eq. (11)) for simplicity. For transportability bias (Figure 1a), let $p_{r=0}^U \neq p_{r=1}^U$ and $p_r^U \sim \mathcal{F}(p)$. Then, these three bias mechanisms are uniquely characterized by the covariance signals (see Definition 4.3) as in Table 2.*

*Proof of Covariance Results for the Transportability Bias Case.* In this case, we have

$$
b_1(X) = \left(p_{r=1}^U - p_{r=0}^U\right)\left(p_{u=1}^Y - p_{u=0}^Y\right), \tag{37}
$$

by Equation (12) in LLemma 4.2. Next,

$$
\begin{aligned}
P(S = 1 \mid X, R = 0) &= p_{u=0}^S + p_{r=0}^U(p_{u=1}^S - p_{u=0}^S) && \text{(Lemma A.2, Eq. (24))} \\
&= p_{u=0}^S. && (38)
\end{aligned}
$$

where the last step is due to $p_{u=0}^S = p_{u=1}^S$ since $S \perp\!\!\!\perp U \mid X, R = 0$ (see Figure 1a). It follows from Eq. (38) that

$$
\text{Var}(S \mid X, R = 0) = p_{u=0}^S(1 - p_{u=0}^S). \tag{39}
$$

Since $p_{r=1}^U$, $p_{r=1}^U$, $p_{u=1}^Y$, $p_{u=0}^Y$, and $p_{u=0}^S$ are independent random variables sampled via Algorithm 1, it follows from Eq. (37) and Eq. (39) that

$$\bar{\rho}(b_1, S) = \mathrm{Cov}(|b_1(X)|, \mathrm{Var}(S|X, R = 0)) = 0. \tag{40}$$

Similarly,

$$
\begin{aligned}
&P(A = 1 \mid X, S = 1, R = 0) \\
&= p_{u=0}^A P(U = 0 \mid X, S = 1, R = 0) + p_{u=1}^A P(U = 1 \mid X, S = 1, R = 0) && \text{(Lemma A.3, Eq. (26))} \\
&= p_{u=0}^A. && (41)
\end{aligned}
$$

where the last step is due to $p_{u=0}^A = p_{u=1}^A$ since $S \perp\!\!\!\perp A \mid X, R = 0$ (see Figure 1a). It follows from Eq. (41) that

$$\mathrm{Var}(A \mid X, S = 1, R = 0) = p_{u=0}^A(1 - p_{u=0}^A). \tag{42}$$

Since $p_{r=1}^U$, $p_{r=1}^U$, $p_{u=1}^Y$, $p_{u=0}^Y$, and $p_{u=0}^A$ are independent random variables, it follows from Eq. (37) and Eq. (42) that

$$\bar{\rho}(b_1, A) = \mathrm{Cov}(|b_1(X)|, \mathrm{Var}(A|X, S = 1, R = 0)) = 0. \tag{43}$$

Next,

$$
\begin{aligned}
&P(Y = 1 \mid X, S = 1, A = 1, R = 0) \\
&= p_{u=0}^Y P(U = 0 \mid X, S = 1, A = 1, R = 0) + p_{u=1}^Y P(U = 1 \mid X, S = 1, A = 1, R = 0) && \text{(Lemma A.4, Eq. (27))} \\
&= p_{u=0}^Y P(U = 0 \mid X, R = 0) + p_{u=1}^Y P(U = 1 \mid X, R = 0) && (44) \\
&= p_{u=0}^Y P(U = 0 \mid R = 0) + p_{u=1}^Y P(U = 1 \mid R = 0) && (45) \\
&= p_{u=0}^Y + p_{r=0}^U(p_{u=1}^Y - p_{u=0}^Y). && (46)
\end{aligned}
$$

where Eq. (44) is due to $S \perp\!\!\!\perp A \perp\!\!\!\perp U \mid X, R = 0$ and Eq. (45) is due to the exogeneity of $X$ and $U$. It follows from Eq. (46) that

$$\mathrm{Var}(Y \mid X, S = 1, A = 1, R = 0) = (p_{u=0}^Y + p_{r=0}^U(p_{u=1}^Y - p_{u=0}^Y))(1 - p_{u=0}^Y - p_{r=0}^U(p_{u=1}^Y - p_{u=0}^Y)). \tag{47}$$

We can then write

$$
\begin{aligned}
\bar{\rho}(b_1, Y) &= \mathrm{Cov}(|b_1(X)|, \mathrm{Var}(Y = 1 \mid X, S = 1, A = 1, R = 0)) \\
&= \mathbb{E}[|b_1(X)|\mathrm{Var}(Y \mid X, S = 1, A = 1, R = 0)] \\
&\quad - \mathbb{E}[|b_1(X)|]\mathbb{E}[\mathrm{Var}(Y \mid X, S = 1, A = 1, R = 0)] \\
&= \int_{\mathcal{F}(p)^4} |x - y||z - w|(y + w(x - y))(1 - y - w(x - y)) \, dP_x \, dP_y \, dP_z \, dP_w \\
&\quad - \int_{\mathcal{F}(p)^4} |x - y||z - w| \, dP_x \, dP_y \, dP_z \, dP_w \\
&\quad \times \int_{\mathcal{F}(p)^3} (y + w(x - y))(1 - y - w(x - y)) \, dP_x \, dP_y \, dP_w
\end{aligned}
\tag{48}
$$

which is nonnegative for all $\mathcal{F}(p) \in \mathcal{F}$ (see Figure 5 for normalized covariance, *i.e.* correlation, $\rho(b_1, Y)$).

Combining Eq. (40), Eq. (43), and Eq. (48) concludes the proof. $\qquad\square$

*Proof of Covariance Results for the Confounding Bias Case.* In this case, we have

$$b_1(X) = \frac{(p_{u=1}^Y - p_{u=0}^Y)(p_{u=1}^A - p_{u=0}^A)}{2(p_{u=1}^A + p_{u=0}^A)}, \tag{49}$$

by Equation (13) in Lemma 4.2. Next,

$$
\begin{aligned}
P(S = 1 \mid X, R = 0) &= p_{u=0}^S + p_{r=0}^U(p_{u=1}^S - p_{u=0}^S) && \text{(Lemma A.2, Eq. (24))} \\
&= p_{u=0}^S. && (50)
\end{aligned}
$$

where the last step is due to $p_{u=0}^S = p_{u=1}^S$ since $S \perp\!\!\!\perp U \mid X, R = 0$ (see Figure 1b). It follows from Eq. (50) that

$$\text{Var}(S \mid X, R = 0) = p_{u=0}^S(1 - p_{u=0}^S). \tag{51}$$

Since $p_{u=1}^A$, $p_{u=0}^A$, $p_{u=1}^Y$, $p_{u=0}^Y$, and $p_{u=0}^S$ are independent random variables sampled via Algorithm 1, it follows from Eq. (49) and Eq. (51) that

$$\bar{\rho}(b_1, S) = \text{Cov}(|b_1(X)|, \text{Var}(S|X, R = 0)) = 0. \tag{52}$$

Next,

$$
\begin{aligned}
&P(A = 1 \mid X, S = 1, R = 0) \\
&= p_{u=0}^A P(U = 0 \mid X, S = 1, R = 0) + p_{u=1}^A P(U = 1 \mid X, S = 1, R = 0) &&\text{(Lemma A.3, Eq. (26))} \\
&= p_{u=0}^A P(U = 0 \mid R = 0) + p_{u=1}^A P(U = 1 \mid R = 0) &&\text{(53)} \\
&= \frac{p_{u=0}^A + p_{u=1}^A}{2} &&\text{(54)}
\end{aligned}
$$

where Eq. (53) follows from $S \perp\!\!\!\perp U \mid X, R = 0$ (see Figure 1b) and the exogeneity of $X$ and $U$, and Eq. (54) from $P(U = 0 \mid R = 0) = 1/2$. It follows from Eq. (54) that

$$\text{Var}(A \mid X, S = 1, R = 0) = \frac{p_{u=0}^A + p_{u=1}^A}{2} \frac{2 - p_{u=0}^A - p_{u=1}^A}{2}. \tag{55}$$

We start with the following key observation: both the bias in Eq. (49) and the conditional variance of $A$ in Eq. (55) are functions of $p_{u=1}^A$ and $p_{u=0}^A$. Hence, their covariance $\bar{\rho}(b_1, A)$ will be nonzero in general. We give an intuitive explanation before the formal result.

Note that $|b_1(X)| \propto |p_{u=1}^A - p_{u=0}^A|$, and that $\text{Var}(A \mid X, S = 1, R = 0)$ is effectively the variance of a Bernoulli random variable with $p = \frac{p_{u=1}^A + p_{u=0}^A}{2}$, which increases toward $p = 0.5$ and decreases toward $p = 0$ and $p = 1$. Since probabilities are confined in the $[0, 1]$ range, $|b_1(X)|$, is maximized when $p_{u=0}^A = 0$ and $p_{u=1}^A = 1$, or vice versa. In those cases, $\frac{p_{u=1}^A + p_{u=0}^A}{2} = 0.5$, that is, $\text{Var}(A \mid X, S = 1, R = 0)$ is also maximized.

In short, $|b_1(X)|$ and $\text{Var}(A \mid X, S = 1, R = 0)$ tend to align. This is intuitive: when $U$'s effect on $A$ is significant, the bias is expected to be larger. Also, not accounting for $U$ induces more uncertainty on the estimated probability of treatment assignment, which is reflected in its variance.

Formally, we are interested in

$$
\begin{aligned}
\bar{\rho}(b_1, A) &= \text{Cov}(|b_1(X)|, \text{Var}(A \mid X, S = 1, R = 0))) \\
&= \mathbb{E}[|b_1(X)|\text{Var}(A \mid X, S = 1, R = 0)] - \mathbb{E}[|b_1(X)|]\mathbb{E}[\text{Var}(A \mid X, S = 1, R = 0)] \\
&= \int_{\mathcal{F}(p)^4} \frac{|x - y||z - w|}{2(z + w)} \frac{z + w}{2} \frac{2 - z - w}{2} \, dPx \, dPy \, dPz \, dPw \\
&\quad - \int_{\mathcal{F}(p)^4} \frac{|x - y||z - w|}{2(z + w)} \, dPx \, dPy \, dPz \, dPw \times \int_{\mathcal{F}(p)^2} \frac{z + w}{2} \frac{(2 - z - w)}{2} \, dPz \, dPw
\end{aligned} \tag{56}
$$

which is nonnegative for all $\mathcal{F}(p) \in \mathcal{F}$ (see Figure 5 for normalized covariance, *i.e.* correlation, $\rho(b_1, A)$).

Next,

$$
\begin{aligned}
&P(Y = 1 \mid X, S = 1, A = 1, R = 0) \\
&= p_{u=0}^Y P(U = 0 \mid X, S = 1, A = 1, R = 0) + p_{u=1}^Y P(U = 1 \mid X, S = 1, A = 1, R = 0) &&\text{(Lemma A.4, Eq. (27))}
\end{aligned}
$$

Here, observe that

$$
\begin{aligned}
&P(U \mid X, S = 1, A = 1, R = 0) \\
&= \frac{P(U, S = 1, A = 1 \mid X, R = 0)}{P(S = 1, A = 1 \mid X, R = 0)} \\
&= \frac{P(U, A = 1 \mid X, R = 0)P(S = 1 \mid X, R = 0)}{P(S = 1 \mid X, R = 0)P(A = 1 \mid X, R = 0)} \quad\quad (S \perp\!\!\!\perp A, U \mid X, R = 0) \\
&= P(U \mid X, A = 1, R = 0) \\
&= \frac{P(A = 1 \mid X, U, R = 0)P(U \mid X, R = 0)}{P(A = 1 \mid X, R = 0)} \\
&= \frac{P(A = 1 \mid X, U, R = 0)P(U \mid R = 0)}{P(A = 1 \mid X, U = 0, R = 0)P(U = 0 \mid R = 0) + P(A = 1 \mid X, U, R = 0)P(U = 1 \mid R = 0)} \quad (X \perp\!\!\!\perp U \mid R = 0) \\
&= \frac{p_U^A}{p_{u=0}^A + p_{u=1}^A} \quad\quad\quad\quad (57)
\end{aligned}
$$

where Eq. (57) is due to $P(U = 0 \mid R = 0) = 1/2$. Plugging Eq. (57) back in we have

$$
P(Y = 1 \mid X, S = 1, A = 1, R = 0) = \frac{p_{u=0}^Y p_{u=0}^A + p_{u=1}^Y p_{u=1}^A}{p_{u=0}^A + p_{u=1}^A} \quad\quad (58)
$$

It follows from Eq. (58) that

$$
\mathrm{Var}(Y \mid X, S = 1, A = 1, R = 0) = \frac{p_{u=0}^Y p_{u=0}^A + p_{u=1}^Y p_{u=1}^A}{p_{u=0}^A + p_{u=1}^A} \left(1 - \frac{p_{u=0}^Y p_{u=0}^A + p_{u=1}^Y p_{u=1}^A}{p_{u=0}^A + p_{u=1}^A}\right). \quad\quad (59)
$$

We can then write

$$
\begin{aligned}
\bar{\rho}(b_1, Y) &= \mathrm{Cov}(|b_1(X)|, \mathrm{Var}(Y = 1 \mid X, S = 1, A = 1, R = 0)) \\
&= \mathbb{E}[|b_1(X)|\mathrm{Var}(Y \mid X, S = 1, A = 1, R = 0)] \\
&\quad - \mathbb{E}[|b_1(X)|]\mathbb{E}[\mathrm{Var}(Y \mid X, S = 1, A = 1, R = 0)] \\
&= \int_{\mathcal{F}(p)^4} \frac{|x - y||z - w|}{2(z + w)} \frac{xz + yw}{z + w}\left(1 - \frac{xz + yw}{z + w}\right) dPx \, dPy \, dPz \, dPw \\
&\quad - \int_{\mathcal{F}(p)^4} \frac{|x - y||z - w|}{2(z + w)} dPx \, dPy \, dPz \, dPw \quad\quad (60) \\
&\quad \times \int_{\mathcal{F}(p)^4} \frac{xz + yw}{z + w}\left(1 - \frac{xz + yw}{z + w}\right) dPx \, dPy \, dPz \, dPw \quad\quad (61)
\end{aligned}
$$

which is nonnegative for all $\mathcal{F}(p) \in \mathcal{F}$ (see Figure 5 for normalized covariance, *i.e.* correlation, $\rho(b_1, Y)$).

Combining Eq. (52), Eq. (56), and Eq. (61) concludes the proof. $\square$

*Proof of Covariance Results for the Selection Bias Case.* In this case, we have

$$
b_1(X) = \frac{\left(p_{u=1}^Y - p_{u=0}^Y\right)\left(p_{u=1}^S - p_{u=0}^S\right)}{2\left(p_{u=1}^S + p_{u=0}^S\right)}, \quad\quad (62)
$$

by Equation (14) in Lemma 4.2. Next,

$$
P(S = 1 \mid X, R = 0) = p_{u=0}^S + p_{r=0}^U(p_{u=1}^S - p_{u=0}^S) \quad\quad \text{(Lemma A.2, Eq. (24))}
$$

$$
= \frac{p_{u=0}^S + p_{u=1}^S}{2}. \quad\quad (63)
$$

where the last step is due to from $p_{r=0}^U = 1/2$. It follows from Eq. (63) that

$$
\mathrm{Var}(S = 1 \mid X, R = 0) = \frac{p_{u=0}^S + p_{u=1}^S}{2} \frac{(2 - p_{u=0}^S - p_{u=1}^S)}{2}. \qu\quad (64)
$$

Notice that Eq. (62) and Eq. (64) follow the same formats as Eq. (49) and Eq. (55), respectively. Then, following the same steps in the derivation of Eq. (56), we have

$$\bar{\rho}(b_1, S) > 0. \tag{65}$$

for all $\mathcal{F}(p) \in \mathcal{F}$.

Next,

$$
\begin{aligned}
&P(A = 1 \mid X, S = 1, R = 0) \\
&= p_{u=0}^A P(U = 0 \mid X, S = 1, R = 0) + p_{u=1}^A P(U = 1 \mid X, S = 1, R = 0) && \text{(Lemma A.3, Eq. (26))} \\
&= p_{u=0}^A P(U = 0 \mid X, S = 1, R = 0) + p_{u=0}^A P(U = 1 \mid X, S = 1, R = 0) && (66) \\
&= p_{u=0}^A && (67)
\end{aligned}
$$

where Eq. (66) follows from $A \perp\!\!\!\perp U \mid X, R = 0$, hence $p_{u=0}^A = p_{u=1}^A$ (see Figure 1c). Eq. (67) follows simply because $P(U = 0 \mid X, S = 1, R = 0) + P(U = 1 \mid X, S = 1, R = 0) = 1$. It follows from Eq. (67) that

$$\text{Var}(A \mid X, S = 1, R = 0) = p_{u=0}^A(1 - p_{u=0}^A). \tag{68}$$

Since $p_{u=1}^S$, $p_{u=0}^S$, $p_{u=1}^Y$, $p_{u=0}^Y$, and $p_{u=0}^A$ are independent random variables sampled via Algorithm 1, it follows from Eq. (62) and Eq. (68) that

$$\bar{\rho}(b_1, A) = \text{Cov}(|b_1(X)|, \text{Var}(S|X, R = 0)) = 0. \tag{69}$$

Next,

$$
\begin{aligned}
&P(Y = 1 \mid X, S = 1, A = 1, R = 0) \\
&= p_{u=0}^Y P(U = 0 \mid X, S = 1, A = 1, R = 0) + p_{u=1}^Y P(U = 1 \mid X, S = 1, A = 1, R = 0) && \text{(Lemma A.4, Eq. (27))}
\end{aligned}
$$

Here, observe that

$$
\begin{aligned}
&P(U \mid X, S = 1, A = 1, R = 0) \\
&= \frac{P(U, S = 1, A = 1 \mid X, R = 0)}{P(S = 1, A = 1 \mid X, R = 0)} \\
&= \frac{P(U, S = 1 \mid X, R = 0)P(A = 1 \mid X, R = 0)}{P(S = 1 \mid X, R = 0)P(A = 1 \mid X, R = 0)} && (A \perp\!\!\!\perp S, U \mid X, R = 0) \\
&= P(U \mid X, S = 1, R = 0) \\
&= \frac{P(S = 1 \mid X, U, R = 0)P(U \mid X, R = 0)}{P(S = 1 \mid X, R = 0)} \\
&= \frac{P(S = 1 \mid X, U, R = 0)P(U \mid R = 0)}{P(S = 1 \mid X, U = 0, R = 0)P(U = 0 \mid R = 0) + P(S = 1 \mid X, U, R = 0)P(U = 1 \mid R = 0)} && (X \perp\!\!\!\perp U \mid R = 0) \\
&= \frac{p_U^S}{p_{u=0}^S + p_{u=1}^S} && (70)
\end{aligned}
$$

where Eq. (70) is due to $P(U = 0 \mid R = 0) = 1/2$. Plugging Eq. (70) back in we have

$$P(Y = 1 \mid X, S = 1, A = 1, R = 0) = \frac{p_{u=0}^Y p_{u=0}^S + p_{u=1}^Y p_{u=1}^S}{p_{u=0}^S + p_{u=1}^S} \tag{71}$$

It follows from Eq. (71) that

$$\text{Var}(Y \mid X, S = 1, A = 1, R = 0) = \frac{p_{u=0}^Y p_{u=0}^S + p_{u=1}^Y p_{u=1}^S}{p_{u=0}^S + p_{u=1}^S}\left(1 - \frac{p_{u=0}^Y p_{u=0}^S + p_{u=1}^Y p_{u=1}^S}{p_{u=0}^S + p_{u=1}^S}\right). \tag{72}$$

Notice that Eq. (62) and Eq. (72) follow the same formats as Eq. (49) and Eq. (59), respectively. Then, following the same steps in the derivation of Eq. (61), we have

$$\bar{\rho}(b_1, Y) > 0. \tag{73}$$

for all $\mathcal{F}(p) \in \mathcal{F}$.

Combining Eq. (65), Eq. (69), and Eq. (73) concludes the proof. $\qquad\square$

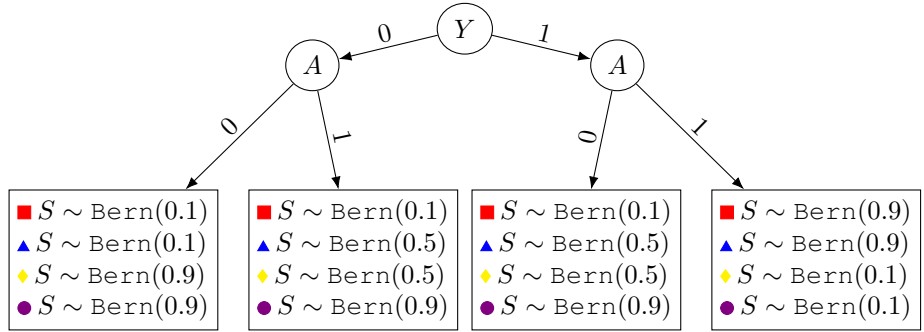

*Figure 6.* Four different specifications of the selection model. See Eq. (85) - (88) for the values of the correlation signals in each case.

### A.3. Proof of the Results in Section 4.4

As alluded to in the main paper, type 2 selection bias requires modified assumptions given its distinct graph structure, i.e. absence of $U$. We repeat Assumption 4.5 here for clarity.

**Assumption A.6.** We have *transportability* $Y^a \perp\!\!\!\perp R|X$; and *ignorability* $Y^a \perp\!\!\!\perp A|X, R$ for $a \in \{0,1\}$.

The covariance signals in this case depend on the specific selection mechanism characterized by $P(S = 1|Y = y, A = a)$. We consider the specifications shown in Figure 6 of the selection model in our synthetic experiments as well as in our theoretical analysis. Conceptually, in our analysis, we show that the sign of the covariance signals differ based on the selection mechanism, implying that the covariances are nonzero, in general. First, we give adjusted definitions that we will use in our derivations.

**Definition A.7.** (Abbreviations for treated group in OS, selection bias type 2)

$$p^A := P(A = 1|X, R = 0) \tag{74}$$

$$p^S_{Y=y,A=a} := P(S = 1|X, A = a, Y = y, R = 0) \tag{75}$$

$$p^Y_a := P(Y^a = 1|X) \tag{76}$$

We need not condition on $A$ and $R$ in (76) due to our assumptions of ignorability and transportability, respectively. Furthermore, we are unable to eliminate the conditioning on $A$ and $Y$ in (75) due to the lack of conditional ignorability in the type 2 selection bias setting.

We can express the bias mechanism, $b_1(X)$, under type 2 selection bias as follows,

**Lemma A.8** (Selection Bias Type 2). *Consider Figure 1d where* $Y^a \not\!\perp\!\!\!\perp A \mid X, S, R = 0$. *Then, we have,*

$$b_1(X) = p^Y_1 \left( 1 - \frac{p^S_{Y=1,A=1}}{p^S_{Y=1,A=1}p^Y_1 + p^S_{Y=0,A=1}(1 - p^Y_1)} \right) \tag{77}$$

*Proof.* As a shorthand, we will define $p^S_{Y=y,A=a} = p^S_{ya}$. We begin by showing expressions for $g_a(X)$ and $f_a(X)$

individually. We have,

$$
\begin{aligned}
g_a(X) &= P(Y^a = 1 \mid X, R = 1) \\
&= \sum_{s \in \{0,1\}} P(Y^a = 1, S = s \mid X, R = 1) \\
&= \sum_{s \in \{0,1\}} P(Y^a = 1 \mid X, S = s, R = 1) P(S = s \mid X, R = 1) \\
&= \sum_{s \in \{0,1\}} P(Y^a = 1 \mid X, R = 1) P(S = s \mid X, R = 1), \qquad \text{given } Y^a \perp\!\!\!\perp S \mid X, R \\
&= \sum_{s \in \{0,1\}} P(Y^a = 1 \mid X) P(S = s \mid X, R = 1), \qquad \text{given } Y^a \perp\!\!\!\perp R \mid X \\
&= P(Y^a = 1 \mid X) \sum_{s \in \{0,1\}} P(S = s \mid X, R = 1) \\
&= P(Y^a = 1 \mid X)
\end{aligned}
$$

Next, we have,

$$
\begin{aligned}
f_a(X) &= P(Y = 1 \mid X, S = 1, A = a, R = 0) \\
&= \frac{P(Y = 1, S = 1 \mid X, A = a, R = 0)}{P(S = 1 \mid X, A = a, R = 0)} \\
&= \frac{P(S = 1 \mid X, Y = 1, A = a, R = 0) P(Y = 1 \mid X, A = a, R = 0)}{P(S = 1 \mid X, A = a, R = 0)} \\
&= \frac{P(S = 1 \mid X, Y = 1, A = a, R = 0) P(Y^a = 1 \mid X, A = a, R = 0)}{P(S = 1 \mid X, A = a, R = 0)} \\
&= \frac{P(S = 1 \mid X, Y = 1, A = a, R = 0) P(Y^a = 1 \mid X, R = 0)}{P(S = 1 \mid X, A = a, R = 0)}, \qquad \text{given } Y^a \perp\!\!\!\perp A \mid X, R \\
&= \frac{P(S = 1 \mid X, Y = 1, A = a, R = 0) P(Y^a = 1 \mid X)}{P(S = 1 \mid X, A = a, R = 0)}, \qquad \text{given } Y^a \perp\!\!\!\perp R \mid X
\end{aligned}
$$

Combining the above two expressions, we have,

$$
\begin{aligned}
b_a(X) &= g_a(X) - f_a(X) \\
&= P(Y^a = 1 \mid X) - \frac{P(S = 1 \mid X, Y = 1, A = a, R = 0) P(Y^a = 1 \mid X)}{P(S = 1 \mid X, A = a, R = 0)} \\
&= P(Y^a = 1 \mid X) \left( 1 - \frac{P(S = 1 \mid X, Y = 1, A = a, R = 0)}{P(S = 1 \mid X, A = a, R = 0)} \right) \\
&= P(Y^a = 1 \mid X) \\
&\quad \left( 1 - \frac{P(S = 1 \mid X, Y = 1, A = a, R = 0)}{\sum_{y \in \{0,1\}} P(S = 1 \mid X, Y = y, A = a, R = 0) P(Y = y \mid X, A = a, R = 0)} \right) \\
&= P(Y^a = 1 \mid X) \\
&\quad \left( 1 - \frac{P(S = 1 \mid X, Y = 1, A = a, R = 0)}{\sum_{y \in \{0,1\}} P(S = 1 \mid X, Y = y, A = a, R = 0) P(Y^a = y \mid X, A = a, R = 0)} \right) \\
&= P(Y^a = 1 \mid X) \left( 1 - \frac{P(S = 1 \mid X, Y = 1, A = a, R = 0)}{\sum_{y \in \{0,1\}} P(S = 1 \mid X, Y = y, A = a, R = 0) P(Y^a = y \mid X)} \right) \\
&= p_a^Y \left( 1 - \frac{p_{1a}^S}{p_{1a}^S p_a^Y + p_{0a}^S (1 - p_a^Y)} \right)
\end{aligned}
$$

completing the proof. $\square$

**Theorem 4.6.** *Suppose that Assumption 4.5 holds and the downstream variables $T \in \{S, A, Y^0, Y^1\}$ are sampled according to Algorithm 1 for some distribution $\mathcal{F}(p) \in \mathcal{F}$. Consider type 2 selection bias in Figure 1d where $Y^a \not\perp\!\!\!\perp A \mid X, S, R = 0$. Then, covariance signals (see Definition 4.3) will be non-zero in general.*

*Proof.* From Lemma A.8, we have,

$$b_1(X) = p_1^Y \left( 1 - \frac{p_{11}^S}{p_{11}^S p_1^Y + p_{01}^S (1 - p_1^Y)} \right)$$

We will compute the variance of each of the target variables, $Y$, $A$, and $S$, followed by the covariance of the bias and the conditional variances. Starting with $Y$, we have,

$$
\begin{aligned}
P(Y &= 1 \mid X, S = 1, A = 1, R = 0) \\
&= \frac{P(Y = 1, S = 1 \mid X, A = 1, R = 0)}{P(S = 1 \mid X, A = 1, R = 0)} \\
&= \frac{P(S = 1 \mid X, Y = 1, A = 1, R = 0)P(Y = 1 \mid X, A = 1, R = 0)}{P(S = 1 \mid X, A = 1, R = 0)} \\
&= \frac{P(S = 1 \mid X, Y = 1, A = 1, R = 0)P(Y^1 = 1 \mid X, A = 1, R = 0)}{P(S = 1 \mid X, A = 1, R = 0)} \\
&= \frac{P(S = 1 \mid X, Y = 1, A = 1, R = 0)P(Y^1 = 1 \mid X)}{P(S = 1 \mid X, A = 1, R = 0)}, \quad \text{by Assmp. 4.5} \\
&= \frac{P(S = 1 \mid X, Y = 1, A = 1, R = 0)P(Y^1 = 1 \mid X)}{\sum_y P(S = 1 \mid X, Y = y, A = 1, R = 0)P(Y = y \mid X, A = 1, R = 0)} \\
&= \frac{P(S = 1 \mid X, Y = 1, A = 1, R = 0)P(Y^1 = 1 \mid X)}{\sum_y P(S = 1 \mid X, Y = y, A = 1, R = 0)P(Y^1 = y \mid X, A = 1, R = 0)} \\
&= \frac{P(S = 1 \mid X, Y = 1, A = 1, R = 0)P(Y^1 = 1 \mid X)}{\sum_y P(S = 1 \mid X, Y = y, A = 1, R = 0)P(Y^1 = y \mid X)}, \quad \text{by Assmp. 4.5} \\
&= \frac{p_{11}^S p_1^Y}{p_{11}^S p_1^Y + p_{01}^S (1 - p_1^Y)}
\end{aligned}
\tag{78}
$$

It follows from Eq. (78) that,

$$\mathrm{Var}(Y \mid X, S = 1, A = 1, R = 0) = \left( \frac{p_{11}^S p_1^Y}{p_{11}^S p_1^Y + p_{01}^S (1 - p_1^Y)} \right) \left( 1 - \frac{p_{11}^S p_1^Y}{p_{11}^S p_1^Y + p_{01}^S (1 - p_1^Y)} \right) \tag{79}$$

Next, we consider $S$. We have,

$$P(S = 1|X, R = 0)$$
$$= \sum_{y,a} P(S = 1 \mid X, Y = y, A = a, R = 0)P(Y = y, A = a|X, R = 0)$$
$$= \sum_{y,a} P(S = 1 \mid X, Y = y, A = a, R = 0)$$
$$P(Y = y \mid X, A = a, R = 0)P(A = a|X, R = 0)$$
$$= \sum_{y,a} P(S = 1 \mid X, Y = y, A = a, R = 0)$$
$$P(Y^a = y \mid X, A = a, R = 0)P(A = a|X, R = 0)$$
$$= \sum_{y,a} P(S = 1 \mid X, Y^a = y, A = a, R = 0)$$
$$P(Y^a = y \mid X)P(A = a|X, R = 0) \quad \text{by Assmp. 4.5}$$
$$= \underbrace{p_{11}^S p_1^Y p^A + p_{10}^S p_0^Y (1 - p^A) + p_{01}^S (1 - p_1^Y)p^A + p_{00}^S (1 - p_0^Y)(1 - p^A)}_{p^{S|X}} \tag{80}$$

It follows from Eq. (80) that,

$$\text{Var}(S \mid X, R = 0) = p^{S|X}(1 - p^{S|X}) \tag{81}$$

Next, we consider $A$. We have,

$$P(S = 1, A = 1 \mid X, R = 0)$$
$$= P(S = 1 \mid X, A = 1, R = 0)P(A = 1|X, R = 0)$$
$$= \sum_y P(S = 1 \mid X, A = 1, Y = y, R = 0)P(Y^1 = y \mid X, A = 1, R = 0)P(A = 1|X, R = 0)$$
$$= \sum_y P(S = 1 \mid X, A = 1, Y = y, R = 0)P(Y^1 = y \mid X)P(A = 1|X, R = 0) \quad \text{by Assmp. 4.5}$$
$$= p_{11}^S p_1^Y p^A + p_{01}^S (1 - p_1^Y)p^A \tag{82}$$

Hence, we have from Eq. (80) and Eq. (82),

$$P(A = 1 \mid X, S = 1, R = 0)$$
$$= \frac{P(S = 1, A = 1 \mid X, R = 0)}{P(S = 1 \mid X, R = 0)}$$
$$= \underbrace{\frac{p^{S_{11}} p^{Y_1} p^A + p^{S_{01}}(1 - p^{Y_1})p^A}{p^{S_{11}} p^{Y_1} p^A + p^{S_{10}} p^{Y_0}(1 - p^A) + p^{S_{01}}(1 - p^{Y_1})p^A + p^{S_{00}}(1 - p^{Y_0})(1 - p^A)}}_{p^{A|X,S}} \tag{83}$$

It follows from Eq. (83) that,

$$\text{Var}(A \mid X, S = 1, R = 0) = p^{A|X,S}(1 - p^{A|X,S}) \tag{84}$$

We compute each covariance signal as in the prior proofs for $T \in \{A, Y, S\}$ and for each of the selection mechanisms specified in Figure 6,

$$\bar{\rho}(b_1, T) = \text{Cov}(|b_1(X)|, \text{Var}(T \mid \cdot))$$
$$= \mathbb{E}[|b_1(X)|\text{Var}(T \mid \cdot)] - \mathbb{E}[|b_1(X)|]\mathbb{E}[\text{Var}(A \mid \cdot)]$$

followed by normalization by the standard deviations of the bias and variance. We get the following results for each selection mechanism,

$$\{\blacksquare P_{00}^S = 0.1, P_{01}^S = 0.1, P_{10}^S = 0.1, P_{11}^S = 0.9\}$$
$$— \rho(b_1, S) = -0.66, \rho(b_1, A) = 0.013, \rho(b_1, Y) = 0.98 \tag{85}$$

$$\{\blacktriangle P_{00}^S = 0.1, P_{01}^S = 0.5, P_{10}^S = 0.5, P_{11}^S = 0.9\}$$
$$— \rho(b_1, S) = 0.33, \rho(b_1, A) = -0.010, \rho(b_1, Y) = 0.95 \tag{86}$$

$$\{\blacklozenge P_{00}^S = 0.9, P_{01}^S = 0.5, P_{10}^S = 0.5, P_{11}^S = 0.1\}$$
$$— \rho(b_1, S) = -0.37, \rho(b_1, A) = 0.058, \rho(b_1, Y) = 0.98 \tag{87}$$

$$\{\bullet P_{00}^S = 0.9, P_{01}^S = 0.9, P_{10}^S = 0.9, P_{11}^S = 0.1\}$$
$$— \rho(b_1, S) = 0.63, \rho(b_1, A) = 0.052, \rho(b_1, Y) = 0.97 \tag{88}$$

The above implies that the covariance signals are nonzero, in general, completing the proof.

$\square$

## A.4. Proofs of the Results in Section 4.5

We repeat our shorthand notation of the covariance signals as well as the definitions of the nuisance function estimators for clarity.

**Definition A.9** (Covariance signals)**.**

$$\bar{\rho}(b_1, S) := \text{Cov}\big(|b_1(X)|, \text{Var}(S | X, R = 0)\big).$$
$$\bar{\rho}(b_1, A) := \text{Cov}\big(|b_1(X)|, \text{Var}(A | X, S = 1, R = 0)\big).$$
$$\bar{\rho}(b_1, Y) := \text{Cov}\big(|b_1(X)|, \text{Var}(Y | X, S = 1, A = 1, R = 0)\big).$$

**Definition A.10** (Nuisance estimators in the OS)**.**

$$\widehat{\eta}_S(X) := \widehat{P}(S = 1 \mid X, R = 0). \tag{15}$$

$$\widehat{\eta}_A(X) := \widehat{P}(A = 1 \mid X, S = 1, R = 0). \tag{16}$$

$$\widehat{\eta}_Y(X) := \widehat{P}(Y = 1 \mid X, S = 1, A = 1, R = 0). \tag{17}$$

$$\widehat{b}_1(X) := \widehat{g}_1(X) - \widehat{f}_1(X), \tag{18}$$

**Lemma A.11.** *Assume that Eq.* (15)*-Eq.* (18) *are strongly consistent estimators, and are dominated by an integrable function* $H(X)$. *Then, for any* $T \in \{S, A, Y\}$, *we have in the limit as* $n \to \infty$,

$$\mathbb{E}_{X,T}\left[|\widehat{b}_1(X)|(T_j - \widehat{\eta}_T(X))^2\right] = \mathbb{E}_X\left[|b_1(X)|\text{Var}_{T|X}(T|X, \cdot)\right], \tag{89}$$

*where* $T \mid X, \cdot \in \{Y \mid X, S = 1, A = 1, R = 0; A \mid X, S = 1, R = 0; S \mid X, R = 0\}$, *for Y, A, and S, respectively.*

*Proof.* For simplicity, we set $T = Y$, since the analysis for $T = \{A, S\}$ is the same. Further, we use an $n$-subscript to make limiting arguments more explicit, where $n$ refers to the estimator fit using $n$-many independent samples in the observational study. Observe,

$$\mathbb{E}_{X,Y}\left[|\widehat{b}_{1,n}(X)|(Y - \widehat{\eta}_{Y,n}(X))^2\right] = \mathbb{E}_X[|\widehat{b}_{1,n}(X)|E_{Y|X}[(Y - \widehat{\eta}_{Y,n}(X))^2]]$$
$$= \mathbb{E}_X\left[|\widehat{b}_{1,n}(X)|\big(\text{Var}_{Y|X}(Y \mid X, \cdot) + (\eta_Y(X) - \widehat{\eta}_{Y,n}(X))^2\big)\right] \tag{90}$$

Here, let us define

$$\widehat{h}_n(X) := |\widehat{b}_{1,n}(X)|\big(\text{Var}_{Y|X}(Y \mid X, \cdot) + (\eta_Y(X) - \widehat{\eta}_{Y,n}(X))^2\big). \tag{91}$$

By the algebra of limits we have

$$\lim_n \widehat{h}_n(X) = \left(\lim_n |\widehat{b}_{1,n}(X)|\right) \times \left(\text{Var}_{Y|X}(Y \mid X, \cdot) + \lim_n (\eta_Y(X) - \widehat{\eta}_{Y,n}(X))^2\right)$$
$$= |b_1(X)|\text{Var}_{Y|X}(Y \mid X, \cdot). \tag{92}$$

almost surely in $X$, due to the strong consistency of the nuisance function estimators and the bias function. That is, $\widehat{h}_n(X)$ is strongly consistent.

To apply the Dominated Convergence Theorem, one also needs $\widehat{h}_n(X)$ to be dominated by an integrable function. This follows after we assume that our nuisance functions estimators are individually dominated by an integrable function $H(X)$. By triangle inequality, we have

$$|\widehat{h}_n(X)| = |\widehat{b}_{1,n}(X)|\text{Var}_{Y|X}(Y \mid X, \cdot) + |\widehat{b}_{1,n}(X)|(\eta_Y(X) - \widehat{\eta}_{Y,n}(X))^2 \tag{93}$$
$$\leq H(X)\text{Var}_{Y|X}(Y \mid X, \cdot) + H(X)(|\eta_Y(X)| + H(X))^2 \tag{94}$$

This upper bound is an integrable function when $H(X)\text{Var}_{Y|X}(Y \mid X, \cdot)$ is integrable and $\eta_Y(X)$ is bounded. Therefore, $\widehat{h}_n(X)$ are dominated by an integrable function for all $n$.

Finally, since $\widehat{h}_n(X)$ is strongly consistent (*i.e.,* almost sure convergence) and is dominated by an integrable function, we can apply the Dominated Convergence Theorem to interchange the limit and expectation:

$$\lim_n \mathbb{E}_X[\widehat{h}_n(X)] = \mathbb{E}_X[\lim_n \widehat{h}_n(X)]$$
$$= \mathbb{E}_X[|b_1(X)|\text{Var}_{Y|X}(Y \mid X, \cdot)] \tag{95}$$

almost surely in $X$, where the last step follows from combining Eq. (90), Eq. (91), and Eq. (92), completing the proof.

$\square$

Consider the estimator of the covariance between the magnitude of the bias and the conditional variance of the target variables,

$$\widehat{\bar{\rho}(b_1, T)} = \frac{n}{n-1}\left(\frac{1}{n}\sum_{i=1}^{n}|\widehat{b}_1(X_i)|(T_i - \widehat{\eta}_T(X_i))^2 - \frac{1}{n^2}\sum_{i=1}^{n}\sum_{j=1}^{n}|\widehat{b}_1(X_i)|(T_j - \widehat{\eta}_T(X_i))^2\right).$$

**Theorem 4.8.** *Assume that the estimators in Eq. (15)-Eq. (18) are strongly consistent, and are dominated by an integrable function $H(X)$. Then, for all $T \in \{S, A, Y\}$, $\widehat{\bar{\rho}(b_1, T)}$ are asymptotically unbiased and strongly consistent estimators of $\bar{\rho}(b_1, T)$.*

*Proof.* For simplicity, we will proceed for $T = Y$ and $\widehat{\eta}_Y(X)$, i.e. the estimator of the outcome function. The analysis is the same for $S$ and $\widehat{\eta}_S(X)$ as well as $A$ and $\widehat{\eta}_A(X)$. We consider the terms of the expectation below separately. We will first show the *asymptotic unbiasedness* of our estimator. Note,

$$\mathbb{E}_{X,Y}\left[\widehat{\bar{\rho}(b_1, Y)}\right]$$

$$= \frac{n}{n-1}\left(\underbrace{\mathbb{E}_{X,Y}\left[\frac{1}{n}\sum_{i=1}^{n}|\widehat{b}_1(X_i)| \cdot (Y_i - \widehat{\eta}_Y(X_i))^2\right]}_{(1)} - \right.$$

$$\left.\underbrace{\mathbb{E}_{X,Y}\left[\frac{1}{n}\sum_{i=1}^{n}|\widehat{b}_1(X_i)| \cdot \frac{1}{n}\sum_{i=1}^{n}(Y_i - \widehat{\eta}_Y(X_i))^2\right]}_{(2)}\right)$$

Consider term (1), where we have,

$$\mathbb{E}_{X,Y}\left[\frac{1}{n}\sum_{i=1}^{n}|\widehat{b}_1(X_i)|\cdot(Y_i-\widehat{\eta}_Y(X_i))^2\right] = \frac{1}{n}\sum_{i=1}^{n}\mathbb{E}_{X,Y}\left[|\widehat{b}_1(X_i)|\cdot(Y_i-\widehat{\eta}_Y(X_i))^2\right]$$

$$= \frac{1}{n}\sum_{i=1}^{n}|\widehat{b}_1(X_i)|\cdot(Y_i-\widehat{\eta}_Y(X_i))^2$$

$$= \mathbb{E}_{X,Y}\left[|\widehat{b}_1(X)|\cdot(Y-\widehat{\eta}_Y(X))^2\right]$$

$$= \mathbb{E}_X\left[|b_1(X)|\cdot\text{Var}(Y|X, S=1, A=1, R=0)\right],$$
by Lemma A.11

Subsequently, we will write $\text{Var}(Y|X, S=1, A=1, R=0)$ as $\text{Var}(Y|X,\cdot)$. For term (2), we have,

$$\mathbb{E}_{X,Y}\left[\frac{1}{n}\sum_{i=1}^{n}|\widehat{b}_1(X_i)|\cdot\frac{1}{n}\sum_{i=1}^{n}(Y_i-\widehat{\eta}_Y(X_i))^2\right]$$

$$= \mathbb{E}_X\left[\frac{1}{n}\sum_{i=1}^{n}|\widehat{b}_1(X_i)|\right]\mathbb{E}_{X,Y}\left[\frac{1}{n}\sum_{i=1}^{n}(Y_i-\widehat{\eta}_Y(X_i))^2\right] + \text{Cov}\left(\frac{1}{n}\sum_{i=1}^{n}|\widehat{b}_1(X_i)|, \frac{1}{n}\sum_{i=1}^{n}(Y_i-\widehat{\eta}_Y(X_i))^2\right)$$

$$= \mathbb{E}_X\left[|\widehat{b}_1(X)|\right]\mathbb{E}_X\left[\text{Var}(Y|X,\cdot)\right] + \text{Cov}\left(\frac{1}{n}\sum_{i=1}^{n}|\widehat{b}_1(X_i)|, \frac{1}{n}\sum_{i=1}^{n}(Y_i-\widehat{\eta}_Y(X_i))^2\right),$$
by Lemma A.11

$$= \mathbb{E}_X[|\widehat{b}_1(X)|]\mathbb{E}_X[\text{Var}(Y|X,\cdot)] - \mathbb{E}\left[\frac{1}{n}\sum_{i=1}^{n}|\widehat{b}_1(X_i)|\right]\mathbb{E}\left[\frac{1}{n}\sum_{i=1}^{n}(Y_i-\widehat{\eta}_Y(X_i))^2\right]$$

$$+ \mathbb{E}\left[\frac{1}{n}\sum_{i=1}^{n}|\widehat{b}_1(X_i)|\cdot\frac{1}{n}\sum_{i=1}^{n}(Y_i-\widehat{\eta}_Y(X_i))^2\right]$$

$$= \mathbb{E}_X[|\widehat{b}_1(X)|]\mathbb{E}_X[\text{Var}(Y|X,\cdot)] - \mathbb{E}_X\left[|\widehat{b}_1(X)|\right]\mathbb{E}_{X,Y}\left[(Y-\widehat{\eta}_Y(X))^2\right]$$

$$+ \mathbb{E}\left[\frac{1}{n}\sum_{i=1}^{n}|\widehat{b}_1(X_i)|\cdot\frac{1}{n}\sum_{i=1}^{n}(Y_i-\widehat{\eta}_Y(X_i))^2\right]$$

$$= \mathbb{E}_X[|\widehat{b}_1(X)|]\mathbb{E}_X[\text{Var}(Y|X,\cdot)] - \mathbb{E}_X[|\widehat{b}_1(X)|]\mathbb{E}_X[\text{Var}(Y|X,\cdot)]$$

$$+ \mathbb{E}\left[\frac{1}{n}\sum_{i=1}^{n}|\widehat{b}_1(X_i)|\cdot\frac{1}{n}\sum_{i=1}^{n}(Y_i-\widehat{\eta}_Y(X_i))^2\right], \text{ by Lemma A.11}$$

$$= \mathbb{E}\left[\frac{1}{n^2}\sum_{i=1}^{n}\sum_{j=1}^{n}|\widehat{b}_1(X_i)|(Y_j-\widehat{\eta}_Y(X_j))^2\right]$$

$$= \frac{1}{n^2}\sum_{i=1}^{n}\sum_{j=1}^{n}|\widehat{b}_1(X_i)|(Y_j-\widehat{\eta}_Y(X_j))^2$$

$$= \frac{1}{n^2}\left(n\mathbb{E}[|\widehat{b}_1(X)|(Y-\widehat{\eta}_Y(X))^2] + (n^2-n)\mathbb{E}[|\widehat{b}_1(X)|]\mathbb{E}[(Y-\widehat{\eta}_Y(X))^2]\right) \tag{96}$$

$$= \frac{1}{n^2}\left(n\mathbb{E}[|b_1(X)|\text{Var}(Y|X,\cdot)] + (n^2-n)\mathbb{E}[|b_1(X)|]\mathbb{E}[\text{Var}(Y|X,\cdot)]\right),$$
by Lemma A.11 and consistency of $\widehat{b}_1$

Eq. (96) is derived from the fact that when $i \neq j$, $|b_1(X_i)|$ and $Y_j - \widehat{\eta}_Y(X_j)$ are independent, which occurs in $n^2 - n$ of the $n^2$ terms. The remaining $n$ terms are dependent, since $i = j$ and they rely on the same $X_i$. We combine the terms to get the

desired result,

$$\mathbb{E}_{X,Y}\left[\overline{\hat{\rho}(b_1,Y)}\right]$$

$$= \frac{n}{n-1}\left(\mathbb{E}\left[|b_1(X)|\text{Var}(Y|X,\cdot)\right] - \frac{1}{n}\mathbb{E}[|b_1(X)|\text{Var}(Y|X,\cdot)] - \frac{n-1}{n}\mathbb{E}[|b_1(X)|]\mathbb{E}[\text{Var}(Y|X,\cdot)]\right)$$

$$= \mathbb{E}\left[|b_1(X)|\text{Var}(Y|X,\cdot)\right] - \mathbb{E}[|b_1(X)|]\mathbb{E}[\text{Var}(Y|X,\cdot)]$$

$$= \text{Cov}(|b_1(X)|, \text{Var}(Y|X,\cdot))$$

$$= \bar{\rho}(b_1,Y).$$

Next, note that our estimator's variance converges to zero since it is a linear combination of sample means of i.i.d. bounded random variables. Asymptotic unbiasedness, which we show above, together with the vanishing variance imply consistency. The result follows in a few lines using Chebyshev's inequality and the triangle rule as follows.

Let $\hat{\theta}_n$ be a sequence of estimators for $\theta$ such that:

$$\lim_{n\to\infty}\mathbb{E}[\hat{\theta}_n] = \theta \quad \text{(asymptotically unbiased)}$$

$$\lim_{n\to\infty}\text{Var}(\hat{\theta}_n) = 0 \quad \text{(vanishing variance)}$$

For any $\varepsilon > 0$, by Chebyshev's inequality:

$$P(|\hat{\theta}_n - \mathbb{E}[\hat{\theta}_n]| \geq \varepsilon) \leq \frac{\text{Var}(\hat{\theta}_n)}{\varepsilon^2}$$

By triangle inequality:

$$|\hat{\theta}_n - \theta| \leq |\hat{\theta}_n - \mathbb{E}[\hat{\theta}_n]| + |\mathbb{E}[\hat{\theta}_n] - \theta|$$

For sufficiently large $n$, by asymptotic unbiasedness, $|\mathbb{E}[\hat{\theta}_n] - \theta| < \frac{\varepsilon}{2}$. Therefore:

$$P(|\hat{\theta}_n - \theta| \geq \varepsilon) \leq P\left(|\hat{\theta}_n - \mathbb{E}[\hat{\theta}_n]| \geq \frac{\varepsilon}{2}\right) \leq \frac{4\cdot\text{Var}(\hat{\theta}_n)}{\varepsilon^2}$$

Taking the limit as $n\to\infty$:

$$\lim_{n\to\infty}P(|\hat{\theta}_n - \theta| \geq \varepsilon) \leq \lim_{n\to\infty}\frac{4\cdot\text{Var}(\hat{\theta}_n)}{\varepsilon^2} = 0$$

which proves $\hat{\theta}_n$ is consistent and we are done. $\square$

### A.5. Additional DAGs for Selection Bias Type 2

See Figure 7 for additional DAGs reflecting type 2 selection bias.

## B. Additional Experimental Results

We ran all experiments on a standard 12-core CPU machine.

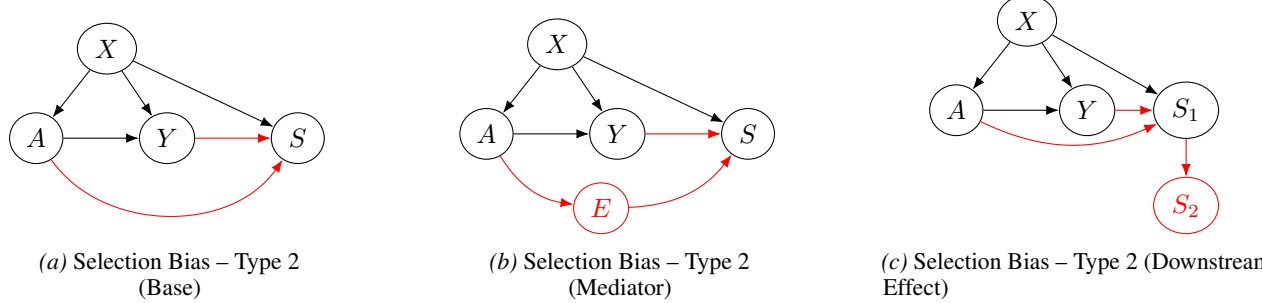

*(a)* Selection Bias – Type 2
(Base)

*(b)* Selection Bias – Type 2
(Mediator)

*(c)* Selection Bias – Type 2 (Downstream
Effect)

*Figure 7.* Common equivalent variants of Selection Bias Type 2.

### B.1. Synthetic Results

**Synthetic settings with varying dimensionalities and selection mechanisms:** We conduct the experiments in Figure 3 with different dimensionalities of the covariates $d \in \{5, 7\}$ ($d = 6$ in the main paper) and RCT sample size $n_{\text{rct}} \in \{2000, 50000\}$. The results are presented in Figure 8. As dimensionality $d$ increases, the power decreases (and type-1 errors increase), *i.e*, it becomes harder to spot the correlation signal statistically significantly. As expected, we observe a similar trend when $n_{\text{rct}}$ decreases, as that leads to larger uncertainty in estimating $\widehat{g}_1(X)$.

We additionally give results for different versions of type 2 selection mechanisms in Figure 9 (see Section A.3 for analytical results). In general, we find that the sign of the correlations changes with the type of selection mechanism, though is always non-zero.

**Synthetic settings with combinations of biases:** In real-world settings, there may be multiple biases present. We experiment with four different combinations of biases: 1) selection bias type 1 and confounding bias, 2) transportability bias and confounding bias, 3) selection bias type 2 and confounding bias, and 4) selection bias type 1 and selection bias type 2. We separately consider the setting with both selection bias type 2 and transportability bias in the next section, Section B.2. Simulation of each bias in a combination is done as described in the main paper in Section 5.

Our experiments, shown in Figure 10, yield several noteworthy insights. Firstly, when transportability bias is combined with confounding bias, the resulting signals can be a blend of the individual effects. Recall that in our analysis, we observed that with only transportability bias, $\rho(b_1, Y) > 0$ and $\rho(b_1, A) = 0$, and with only confounding bias, $\rho(b_1, Y) > 0$ and $\rho(b_1, A) > 0$. Consequently, when both biases are present, we expect $\rho(b_1, A)$ to be somewhat reduced compared to the confounding-only case. This effect is reflected in our results, where the percentage of runs showing statistically significant correlations in the biased region drops from 96% (single bias) to 48% (combined biases).

---

**Algorithm 2:** Generative Model in the OS

---

**Input:** $T \in \{S, A, Y^0, Y^1\}$; Boolean $U$-**bias**;
Bernoulli Parameter Distribution $\mathcal{F}$.

Let $p_{x,u}^T \coloneqq P(T{=}1 | X{=}x, U{=}u, R{=}0)$
**for** $x \in \mathcal{X}$ **do**
    $u \sim \texttt{Unif}(0, 1)$
    **if** $U$-bias **then**
        $p_1 \sim \mathcal{F}, p_0 \sim \mathcal{F}$
        $p_{x,u}^T = u \cdot p_1 + (1 - u) \cdot p_0$
    **else**
        $p_{x,u}^T \sim \mathcal{F}$
    **end if**
**end for**

---

Secondly, we find that the effect of having only selection bias type 2 on the correlation signals is relatively indistinguishable from the effect when selection bias type 2 is combined with another bias (such as selection bias type 1 or confounding bias). Finally, when selection bias type 1 is combined with confounding bias, all examined correlations turn positive, i.e. $\rho(b_1, S) > 0, \rho(b_1, A) > 0, \rho(b_1, Y) > 0$. This is a natural result given that there is an unobserved covariate, $U$, influencing $A, Y$ and $S$.

**Synthetic settings with continuous $U$:** In the main paper, we assume a binary, unobserved confounder $U$ that captures residual effects of multiple unmeasured confounders, primarily to simplify the theoretical analysis. Here, we relax this assumption and extend our synthetic experiments to allow for a continuous $U$. The revised generative model is shown in Algorithm 2. Intuitively, when unobserved confounding is present in a patient subgroup (indicated by the $U$-bias flag), the

probability of the target variable becomes a convex combination of two samples from $\mathcal{F}$, weighted by $U \sim \texttt{Unif(0,1)}$.

We retain the same parameters as in the main paper: $X \in \{0,1\}^d$ with $d = 6$, $n_{\text{rct}} = 50000$, $n_{\text{os}} = 50000$, and $n_{\text{val}} = 2000$. The resulting covariance signals, shown in Figure 11, align with our theoretical predictions, offering further empirical evidence that our framework extends to settings with continuous unobserved confounding.

### B.2. WHI Results

**Type 2 selection bias and transportability bias:** To test the hypothesis that residual transportability bias in the WHI experiments yields an amplified $\rho(b_1, Y)$ signal after correcting for the type 2 selection bias, we conduct a synthetic experiment that incorporates both type 2 selection bias and transportability bias to replicate the WHI setting.

For the type 2 selection bias[4], we experiment with two different configurations: 1) $p_{00}^S = 0.1, p_{01}^S = 0.1, p_{10}^S = 0.1, p_{11}^S = 0.9$ and 2) $p_{00}^S = 0.9, p_{01}^S = 0.9, p_{10}^S = 0.3, p_{11}^S = 0.1$. One example of the former configuration is Berkson's bias, where people who get exposed to the treatment and have the outcome, e.g. hospitalization, are more likely to be selected. The latter configuration is a rough approximation of the selection mechanism in the WHI OS. Specifically, it is designed to mimic a scenario where patients who are treated and subsequently experience an adverse event are more likely to be filtered out, while patients who either were treated without adverse outcomes or did not receive treatment are preferentially selected.

Following the approach used in our previous experiments, we generate plots that show pairs of Pearson correlation coefficients and their corresponding $p$-values for each run. These plots indicate the percentage of runs where the observed correlation coefficients match the predictions outlined in Table 2 and Equation (85) (detailed in Section A.3), using a significance threshold of $p < 0.01$. We repeat this experiment under the alternative selection probabilities (second row) as well as under conditions designed to minimize selection bias by increasing the selection probabilities to 0.99 (third row). In all scenarios, we calculate the average correlation for runs falling within the positive or negative region depending on signal, e.g. negative region for $\rho(b_1, A)$), with error bars representing standard deviation across these runs. The complete results are shown in Figure 12. Importantly, we find that $\rho(b_1, Y)$ tends to increase when minimizing the type 2 selection bias regardless of the selection probabilities, supporting our hypothesis.

**Testing distributional assumption in the WHI data:** One important question is how well our theoretical results apply to a real-world dataset like WHI, given that our analysis relies on a specific generative model that may not perfectly capture reality. To further justify our generative model, we perform a simple analysis to test our assumption about the underlying sampling distribution in our real-world experiment–specifically the feasibility of modeling the conditional distributions of downstream variables, such as $p(A \mid X, U)$, with a low-uncertainty distribution, $\mathcal{F}(p) \sim \mathcal{F}$ (see Equation (7)).

As shown in Figure 13, we plot $P(A \mid X)$ conditioning on different "sequences" of subgroups after starting initially with the marginal distribution in the overall population. These subgroups are LLM-generated with a prompt focused towards subgroups that confer an increased or decreased risk for CHD. We use Claude-3.7-Sonnet to generate subgroup sequences. By way of example, the probability of treatment assignment is 33% for the overall WHI observational cohort, relatively close to 50%. In the age-based subgroup sequence, the uncertainty in treatment assignment decreases, and the empirical probability of treatment assignment drops. For instance, when first conditioning on age $> 65$, $P(A = 1 \mid X)$ decreases from 33% to 22%, and then decreases further when conditioning on both age $> 65$ and elevated BMI. In the case where the subgroups are protective against CHD (bottom row of Figure 13, we see that the probability increases towards the "higher" region of certainty (i.e. $P(A \mid X) > 0.5$). These patterns show that adding covariates reflecting relevant clinical subgroups reduces uncertainty in treatment assignment, albeit in different directions depending on the baseline risk (for the outcome) in the subgroup.

**Additional details for WHI experiments:** In Table 3, we report the average hazard ratios measuring the effect of combined HRT on coronary heart disease (CHD) and stroke outcomes in post-menopausal women.

For the nuisance function estimators, we use logistic regression for $\widehat{\eta}_S$ and $\widehat{\eta}_A$, and random forest models for $\widehat{\eta}_Y$ (see Eq. (15)-Eq. (17)). We use the default hyperparameters in the scikit-learn implementation of the logistic regression model with maximum iterations of 1000 (Buitinck et al., 2013). For the scikit-learn implementation of the random forest, we tune the following hyperparameters on a subset of the training set:

---

[4]for ease of notation, we denote $P(S = 1 \mid Y = y, A = a)$ as $p_{ya}^S$

*Table 3.* Hazard ratios averaged over population in RCT and OS. We consider "baseline" and "corrected" settings in OS.

| Estrogen plus progestin | CHD (Baseline) | | CHD (Corrected) | | Stroke (Baseline) | | Stroke (Corrected) | |
|---|---|---|---|---|---|---|---|---|
| | HR | 95% CI | HR | 95% CI | HR | 95% CI | HR | 95% CI |
| RCT | 1.28 | (1.01,1.61) | — | — | 1.37 | (1.04, 1.8) | — | — |
| OS | 0.87 | (0.73,1.03) | 1.09 | (0.81, 1.45) | 0.86 | (0.71,1.04) | 1.29 | (0.95,1.75) |

- n-estimators: [100,500]

- max-depth: [None, 10, 20]

- min-samples-split: [2,10]

## C. Related Work

### C.1. Latent variable models

In the context of latent variable models and graphical models more generally, the main problems, at least relevant to our setting, of interest are parameter estimation. Using method of moments for parameter estimation in latent variable models is a common technique. The method of moments paradigm involves 1) computation of certain statistics of the observed data, i.e. means and correlations and 2) finding the model parameters that give rise (approximately) to the same corresponding population quantities. One example of how these operations are executed is through tensor decomposition techniques, as in Anandkumar et al. (2014). At a high level, these methods store low-order moments of the observable data in multidimensional tensors; then, the parameters of the desired model are recovered using tensor decomposition. Anandkumar et al. (2011) assume a causal tree structure, where hidden variables serve as internal nodes and observed variables as leaves. They analyze correlations between observed variables (often through singular values of second moment matrices) to infer relationships through latent variables. See also Ruffini et al. (2018), who give a nice overview of these kind of methods.

### C.2. Causal Structure Learning

Our work is closely related to the line of work whose goal is to learn a causal structure from observational data and (potentially) experimental data. Our method differs from these approaches in a crucial way. Rather than identifying exact causal DAGs, we focus on categorizing types of bias, where multiple DAGs may be equivalent. There are two broad classes of approaches for structure learning: constraint-based methods and score-based methods. Below we provide an overview.

**Score-based methods**   The general idea in this line of works is to assign a score to each potential graph, which reflects how well it explains the observed data (Ng et al., 2022; Chobtham & Constantinou, 2020). For instance, Ng et al. (2022) utilize a scoring function based on the degrees of freedom of a graph (i.e. number of edges in the graph) measuring how well a particular graph explains the empirical covariance matrix derived from the observed variables.

**Constraint-based methods**   This line of work uses conditional independence tests of the observed data to infer the underlying causal structure. Some approaches assume causal sufficiency, i.e., assume no selection variables and no unmeasured common causes (e.g., PC, CCD algorithms), while others simply assume the graph is acyclic (e.g., FCI and RFCI algorithms). See Colombo et al. (2014); Akbari et al. (2021); Sadeghi & Soo (2022) for examples of this approach. The main drawback with these methods in our setting is that the CI-based tests will result in potentially multiple graphs that are Markov equivalent (and there is no way to tell if implied relationships between observed variables are due to e.g. a hidden confounder or selection bias). In contrast, our method can allow practitioners to make this distinction.

### C.3. Distribution Shifts

Our work is similar in spirit to some papers in the literature on distribution shifts but differs in its goals. For instance, Cai et al. (2023) investigate how much of the decline in prediction performance can be attributed to covariate shift. Similarly, Jin et al. (2023) attempt to diagnose discrepancies between different studies by breaking them down into components such as sampling variability, observable distribution shifts, and residual factors. However, while their work focuses on quantifying

how much of the discrepancy between studies is due to distribution shifts in observed covariates—a specific form of selection bias—we take a broader approach by examining several potential biases, particularly those relevant to the medical domain.

## C.4. Integrating Evidence from Across Experimental and Observational Studies

Combining information from RCTs and OSes to develop causally reliable and statistically powerful methods has been of great interest recently. Target trial emulation (TTE), for instance, is a commonly adopted framework to analyze observational data in a principled way (Hernán & Robins, 2016; Franklin et al., 2021; Hernán et al., 2022; Wang et al., 2023).

There is a growing body of work focused on integrating observational and experimental data to enable more reliable, generalizable, and statistically efficient causal inference (Bareinboim & Pearl, 2016; NICE, 2022; Colnet et al., 2024; De Bartolomeis et al., 2024a; Boughdiri et al., 2025). Several studies investigate how potentially biased outcome models—learned from observational data—can enhance the statistical power of randomized controlled trials (RCTs) and support the generalization of findings to *target* populations or novel experimental settings (Kallus et al., 2018; Schuler et al., 2022; Hatt et al., 2022; Demirel et al., 2024a; Cadei et al., 2025). Along similar lines, Kaul & Gordon (2025) leverage observational data to construct an *untrusted prior*, which is then combined with experimental data via *conformal prediction* to yield valid and tighter confidence intervals for treatment effects. Guo et al. (2022) propose using observational data to construct control variates, reducing variance in ATE estimation within RCTs. Similarly, Wang et al. (2025) study the derivation of tighter confidence intervals for the ATE when multiple OSes are available. Lastly, Oberst et al. (2023); Rosenman (2025) offer a comprehensive overview of methods that adaptively combine ATE estimates from both experimental and observational data to produce improved hybrid estimators.

## C.5. Benchmarking Observational Studies

Another line of work focuses on *benchmarking*, which involves comparing treatment effect estimates from observational studies (OSes) to those from randomized controlled trials (RCTs) (Hartman et al., 2015; Dahabreh et al., 2020). Benchmarking methods play a crucial role in assessing the reliability of OSes before they are used in downstream decision-making. Moreover, systematic reviews of benchmarking efforts can offer broader insights into the practical strengths and limitations of OS-based analyses (Forbes & Dahabreh, 2020; Wang et al., 2023).

De Bartolomeis et al. (2024a;b) quantify the bias in an OS by directly comparing its estimates to those from an RCT. Rather than focusing on population-level comparisons, Hussain et al. (2022) assess ATE estimates across various subgroups in both studies. Expanding on this approach, Hussain et al. (2023); Demirel et al. (2024b) compare *conditional* ATEs, while enabling *automatic* identification of patient subgroups where the OS and RCT estimates diverge. Other work sidesteps the need for individual-level RCT data entirely: Karlsson & Krijthe (2024; 2025); Xiao et al. (2024) show that hidden confounding can be detected—and in some cases mitigated—using multiple OSes under alternative assumptions. Finally, Fawkes et al. (2025) establish fundamental limits on the extent to which RCTs can be used to falsify or validate observational findings.

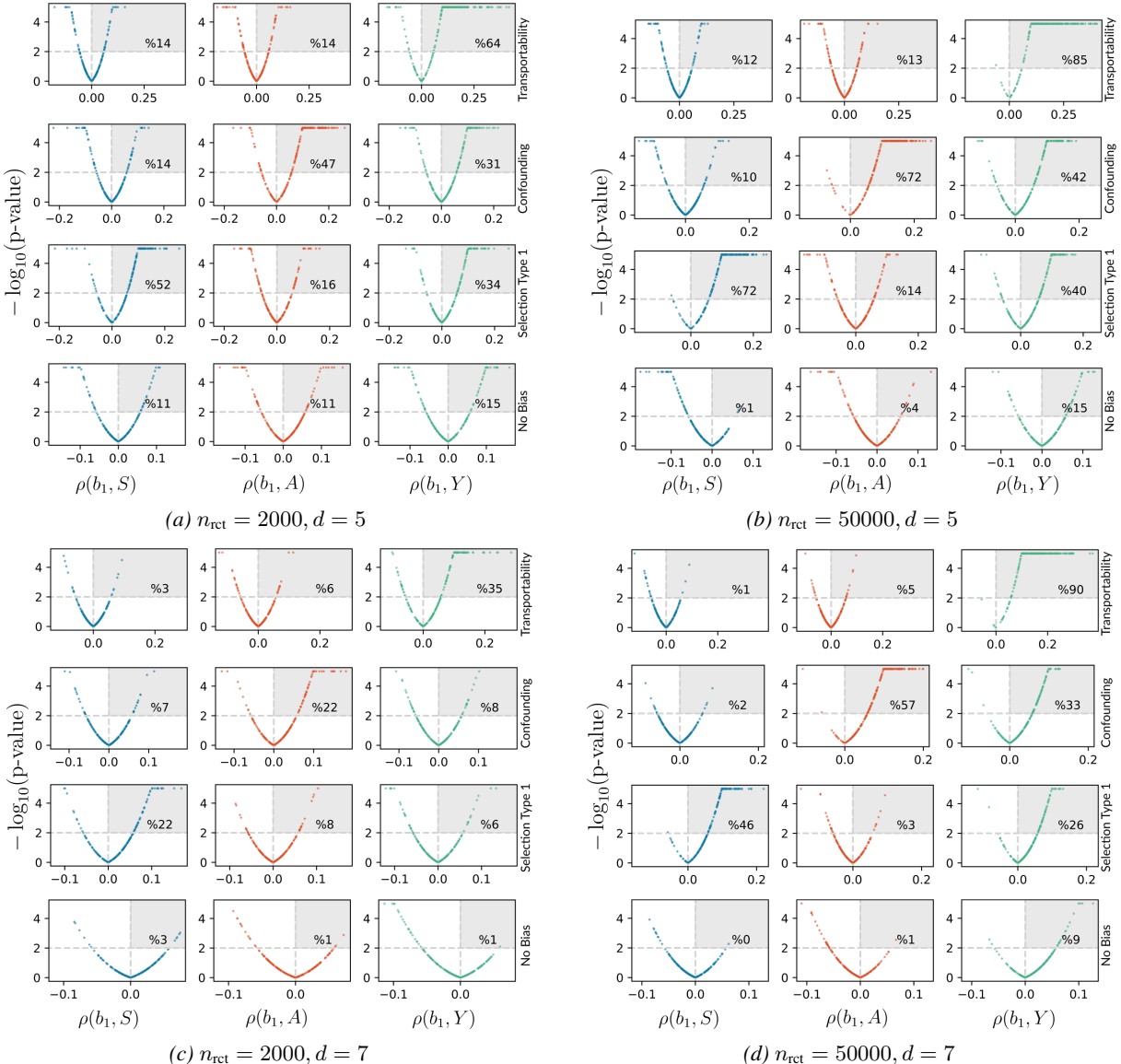

*Figure 8.* Covariance signals for synthetic experiments with different covariate dimensionalities $d$ and RCT sample size $n_{\mathrm{rct}}$.

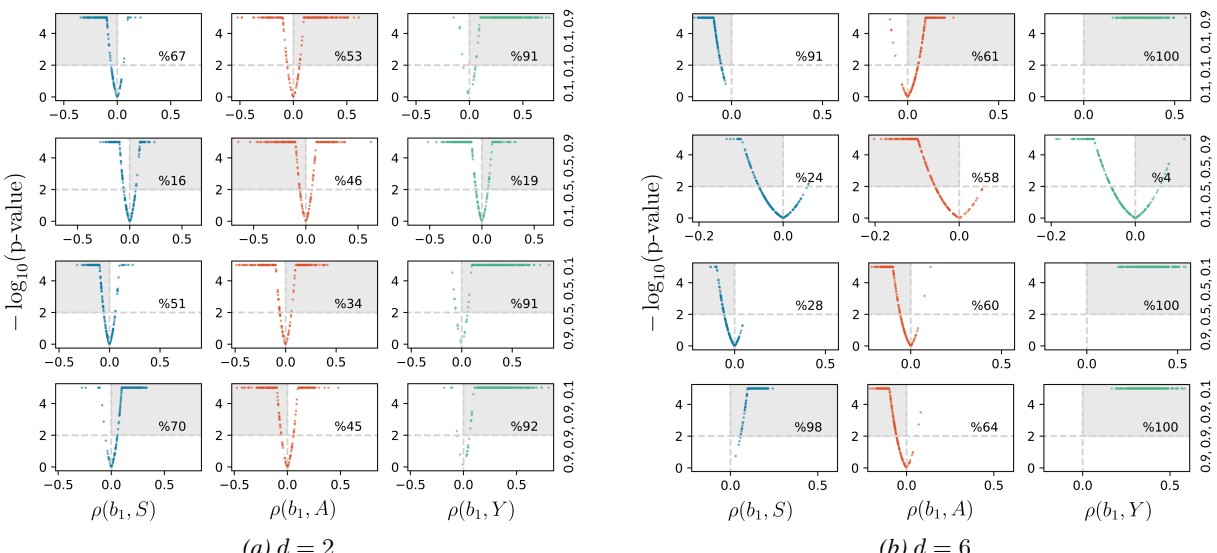

*Figure 9.* Covariance signals for synthetic experiments with different selection mechanisms and covariate dimensionalities $d$.

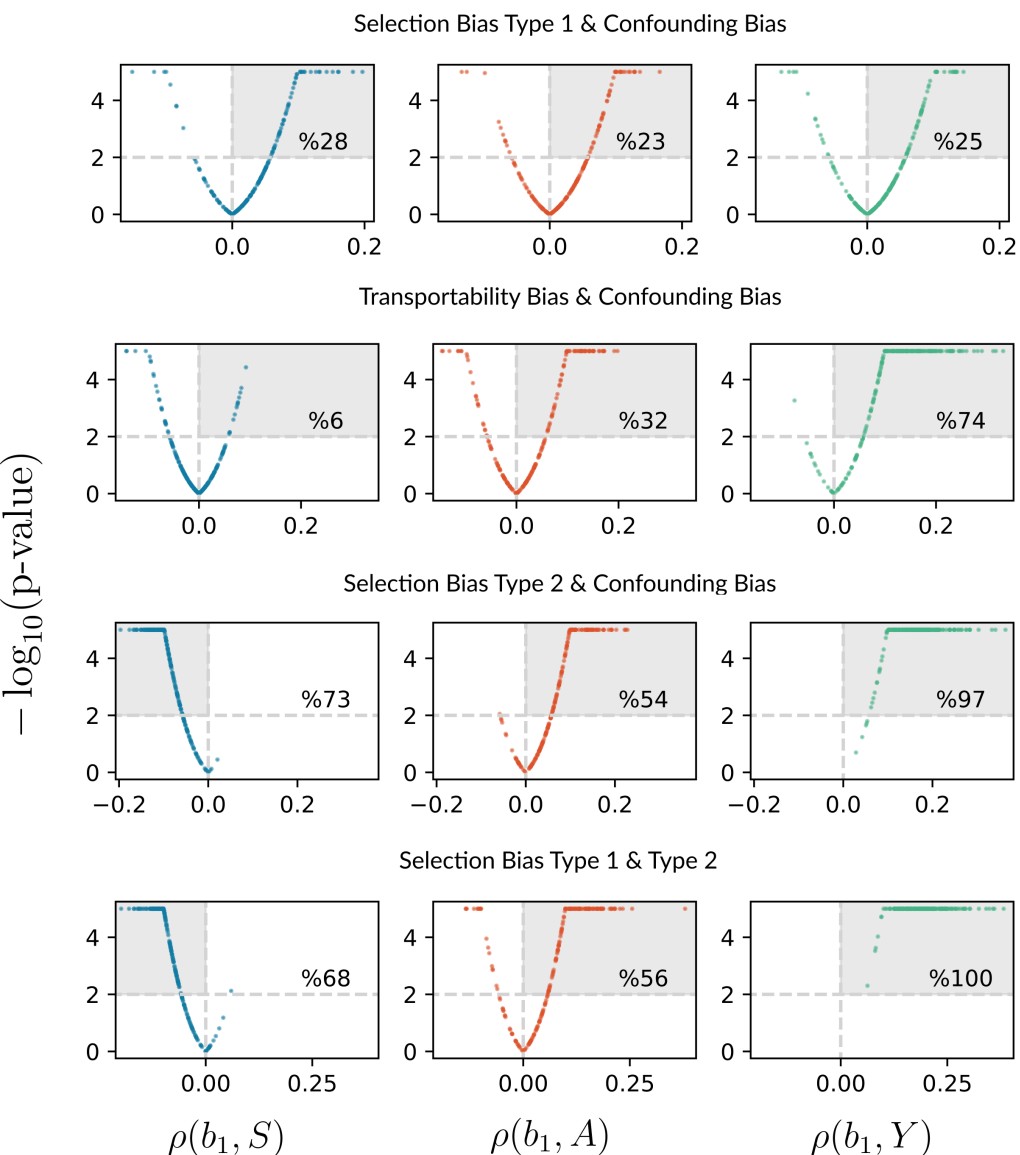

*Figure 10.* Covariance signals for synthetic settings involving different combinations of biases.

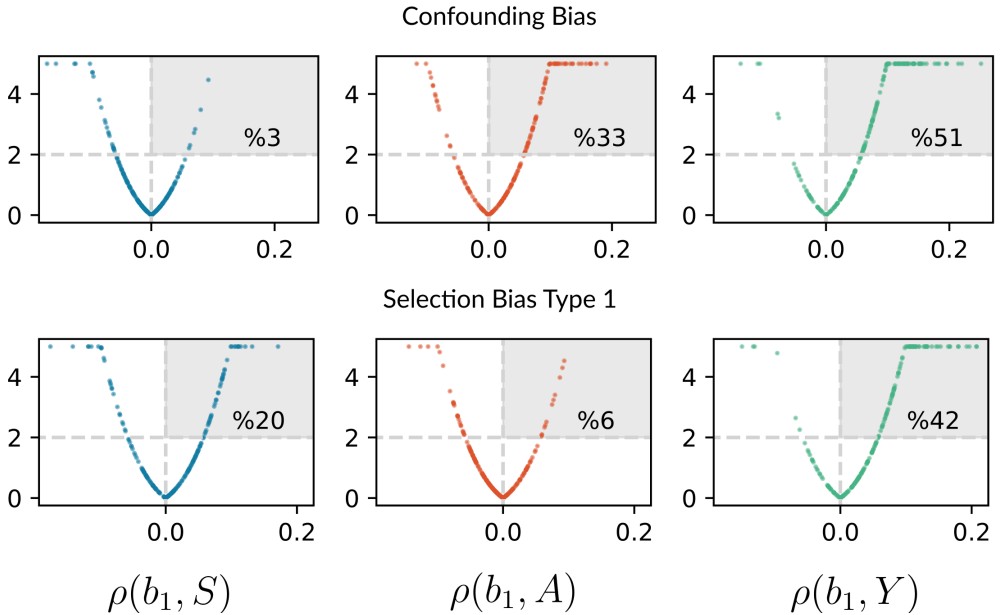

*Figure 11.* Covariance signals for synthetic setting with continuous $U$.

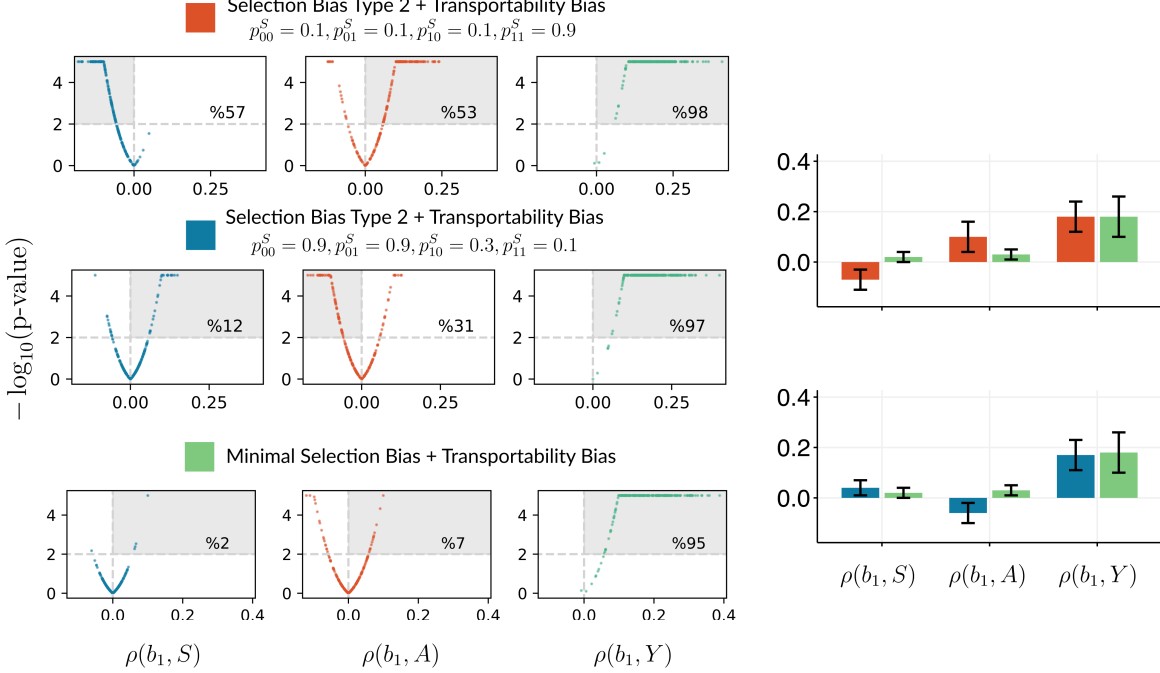

*Figure 12.* Left: Covariance signals under different synthetic settings: top two have type 2 selection bias (with differing selection probabilities) and transportability bias, bottom has minimal selection bias. Right: Average correlation over runs in positive or negative region depending on signal.

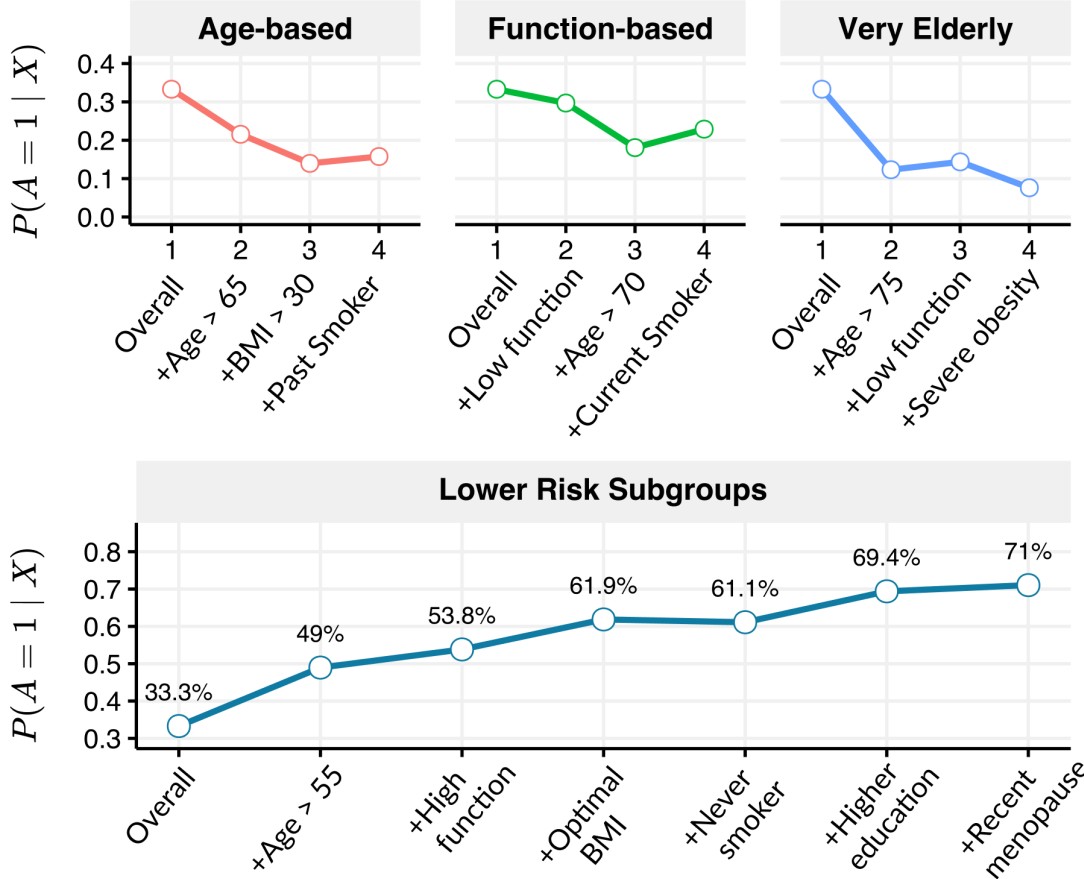

*Figure 13.* Probability of giving HRT in different sequences of subgroups. On the $x$-axis, we stratify into a specific patient subgroup by adding a constraint on a covariate. Top: Patient subgroups are higher-risk groups for CHD and stroke. Bottom: Patient subgroups are lower-risk groups for CHD and stroke.

