# OpenReview forum: "Uncovering Bias Mechanisms in Observational Studies"
_ICML.cc/2026/Conference — ICML 2026 regular_

### Official Review · Reviewer_zHNc · 2026-02-23

**Soundness:** 4
**Presentation:** 4
**Significance:** 3
**Originality:** 3
**Overall Recommendation:** 5
**Confidence:** 4

**Summary:**

Randomised controlled trials are the gold standard for estimating the conditional average treatment effect (CATE) of some intervention. However, we often have only access to observational data, which may introduce bias into CATE estimation. This paper introduces a theoretical causal taxonomy of common biases and a method for identifying which bias is present from observed data. The authors extensively discuss the assumptions necessary for their bias identification method and provide illustrative synthetic experiments. Finally, the method is tested on the real-world WHI dataset.

**Compliance With Llm Reviewing Policy:**

Affirmed.

**Final Justification:**

My final recommendation is 'accept'. I think this is a solid paper that will be interesting to the community. This aligns with the general consensus of the other reviewers.

**Key Questions For Authors:**

>[Line 262 RHS] "One can formalize this idea through the covariances of the absolute bias function and conditional variance functions..."

1. I agree that dependence between these functions could be a useful signal, and am broadly fine with this formulation, however, since covariance is a linear measure of dependence, could this be a limitation of the method? Have you tried other ways of measuring dependence?

**Limitations:**

The authors discuss limitations very well. I think Table 1 in particular provides outstanding clarity in its analysis of assumptions.

**Strengths And Weaknesses:**

I would like to thank the authors for submitting their work, which I found clear and instructive. I think this paper is technically strong and well presented (above the bar for publication at ICML). My three potential concerns are with experimental soundness, relevance, and originality, which I discuss below. Overall, I have recommended weak accept, but am willing to improve my score if the authors/AC/other reviewers can reassure me in the rebuttal phase.

### Soundness

I break down my assessment into theoretical and experimental soundness below. I think that the experimental soundness is potentially a weak part of the paper, however, I appreciate this is not the main point of the work. I have thus rated overall soundness as 'excellent' due to the strong theoretical side.

- Theoretical soundness. The taxonomy of biases is reasonable and well-motivated by examples; the theory in Section 4 is also correct to the best of my understanding. The discussion of assumptions in Table 1 is outstanding. I have not checked the proofs in the Appendix line-by-line, but I have no reason to doubt their soundness.

- Empirical soundness. I think the empirical evaluation is potentially a weak part of the paper. The synthetic experiments are nice to have, however, it's trivial to design synthetic settings which agree with theoretical analysis (since we can always ensure that our assumptions are satisfied). It's good to see that there's a real world experiment, however the results seem much less conclusive than the synthetic ones.

### Presentation

I have rated presentation as 'excellent'. The writing and figures are clear, and the notation and mathematics are well explained. I have two minor suggestions:

- Each time Assumption 2.1 is cross-referenced (e.g. on line 103), it is mislabelled as 'Theorem 2.1'. This is probably just a `cleverref` LaTeX error, but I'd recommend manually fixing it as it tripped me up when reading the paper.

- Figure 4 is quite small and difficult to read without zooming in. Is there a way you can alter the layout to make it larger and easier to read?

### Significance

The topic of bias identification in observational studies is both interesting and, to the best of my knowledge, important. My only concern regarding significance is the relevance for the main conference track at ICML. I have two questions for the authors/AC:

1.  Given how the paper introduces a theoretical framework (and is somewhat weaker on the experimental side), would it would be more suited as a position paper?
1. The paper focuses on CATE estimation from tabular data. As far as I can tell, all models are implemented as either linear/logistic regression/random forests. Much of the ICML community is concerned with deep models and/or high-dimensional data, so I wonder if this will be relevant to the community?

I have thus rated significance as 'good', but I will be more than happy to upgrade my rating if the other reviewers do not share the same concerns about relevance.

### Originality

I am somewhat unsure about how to rate the originality of this paper.

On one hand, causal taxonomies of bias have been studied before (e.g. in fairness [1,2], distribution shift [3], and medical imaging [4], and many more). Similarly, the notion of bias identification (as opposed to mere detection) has been studied before (e.g. for distribution shifts [5]) and the method of using downstream variances to identify the bias seems at least superficially similar to common methods in causal structure learning.

On the other hand, this paper focuses on identifying bias in CATE estimation from tabular data, which is a distinct application, and thus this work may be novel when applied in this setting. Since I am not too familiar with the related work in this exact application, I have rated the overall novelty as 'good'. However, I will read the author responses and other reviews and will adjust my score accordingly.

[1] Plečko, Drago, and Elias Bareinboim. "Causal Fairness Analysis:: A Causal Toolkit for Fair Machine Learning." Foundations and Trends® in Machine Learning 17.3 (2024): 304-589.

[2] Schrouff, Jessica, et al. "Mind the graph when balancing data for fairness or robustness." Advances in Neural Information Processing Systems 37 (2024): 29913-29947.

[3] Veitch, Victor, et al. "Counterfactual invariance to spurious correlations in text classification." Advances in neural information processing systems 34 (2021): 16196-16208.

[4] Castro, Daniel C., Ian Walker, and Ben Glocker. "Causality matters in medical imaging." Nature Communications 11.1 (2020): 3673.

[5] Roschewitz, Mélanie, et al. "Automatic dataset shift identification to support safe deployment of medical imaging AI." International Conference on Medical Image Computing and Computer-Assisted Intervention. Cham: Springer Nature Switzerland, 2025.

---

> ### Author Rebuttal · Authors · 2026-03-30
>
> We thank the reviewer for the thoughtful and encouraging assessment with excellent scores on soundness and presentation. We are especially glad that they found the theory technically sound and interesting, and Table 1 outstandingly helpful. We address the main concerns below.
>
> ## Reviewer Comment #1
>
> > “the synthetic experiments are nice to have, however, it’s trivial to design synthetic settings which agree with theoretical analysis”
>
> We agree that synthetic results alone cannot establish real-world applicability, but that is not their only role in the paper. They are also used to study how sensitive our analytical results are to departures from the assumptions we make for theoretical tractability. In particular, we test whether the main qualitative and quantitative patterns predicted by the theory continue to hold when key assumptions are weakened or violated.
>
> Concretely, beyond the main synthetic experiments in Section 5 and Figure 3, Appendix B.1 examines varying covariate dimensionality and RCT size (Figures 8 and 9), alternative type-2 selection mechanisms (Figure 9), mixtures of multiple bias types (Figure 10), and continuous rather than binary $U$ (Figure 11). We also use synthetic experiments to support our interpretation of the real-world results (kindly see the next point).
>
> Together, these experiments show that the analytical insights are not confined to the narrowest setting in which we develop theory, but remain informative under several important and practically relevant deviations summarized in Table 1. We will make this role of the synthetic experiments clearer in the revision.
>
> ## Reviewer Comment #2
>
> > “the real world experiment … seems much less conclusive than the synthetic ones”
>
> We agree that the real-world results are more nuanced and less conclusive than the synthetic ones. In the WHI analysis, we use our framework to reason about the observed signals in a noisy and realistic setting, and we explicitly try to provide plausible explanations for those signals in the main paper. In Section 6, for example, we interpret the baseline WHI pattern as consistent with type-2 selection bias, and then discuss why the persistence of $\rho(b_1, Y)$ after correction may point to residual transportability bias.
>
> Importantly, we examine this explanation more carefully in Appendix B.2 by running targeted synthetic experiments designed to mimic the WHI setting and test whether our proposed hypothesis for explaining the observed signals reproduce the same signal patterns in a controlled synthetic setting. Kindly see the paragraph *Type 2 selection bias and transportability bias* and Figure 12. We will make this illustrative role of the real-world study as to how one can interpret the signals in complicated real-world setting much clearer in the revision.
>
> ## Reviewer Comment #3
>
> > “would it be more suited as a position paper?” / “I wonder if this will be relevant to the ICML community?”
>
> We believe the paper is a good fit for the main conference because it goes beyond a position statement. It formalizes a problem, proves identification results, develops a consistent estimator, and studies the method in both synthetic and real data. More broadly, although our experiments are in causal inference, the core question is an ML one: whether predictive-performance signals can be used to diagnose failure modes of systems.
>
> ## Reviewer Comment #4
>
> > “causal taxonomies of bias have been studied before … similarly for bias identification … and [this] seems superficially similar to causal structure learning”
>
> We agree that our work is related to these lines of research, and we will sharpen this positioning in the related-work discussion. Our main claim is not that taxonomies or bias identification are new in the abstract, but that our paper introduces a new formulation and solution for identifying bias mechanisms in CATE benchmarking between an RCT and an OS using the covariance between estimated bias and nuisance prediction error. This differs from prior work on fairness or generic distribution shift, and also from causal structure learning: we do not attempt to recover an exact DAG, but instead identify a coarser and more actionable equivalence class of mechanisms.
>
> ## Reviewer Comment #5
>
> > “since covariance is a linear measure of dependence, could this be a limitation?”
>
> We chose covariance because it yields analytically tractable signatures, interpretable estimators, and the identification results in our theory. We do not claim that it is the only or universally best dependence measure, and exploring nonlinear alternatives is an interesting direction for future work. We will clarify this point in the revision.
>
> Finally, thank you for the presentation suggestions. We will fix the incorrect references to Assumptions and Definition, which were indeed caused by our incorrect use of the `cleveref`  package. We are happy to address any remaining questions that can help change your score to a more positive one!

---

> > ### Author Rebuttal · Reviewer_zHNc · 2026-04-01
> >
> > Thanks for taking the time to answer my questions. Overall, I think this is a good submission and will update my score from 4 to 5.

---

> > > ### Author Response · Authors · 2026-04-02
> > >
> > > We greatly appreciate the reviewer's engagement with our rebuttal and updating their score.

---

### Official Review · Reviewer_HgMk · 2026-03-10

**Soundness:** 2
**Presentation:** 3
**Significance:** 2
**Originality:** 3
**Overall Recommendation:** 4
**Confidence:** 3

**Summary:**

The authors propose a technique that can be used to detect different forms of bias that arise in causal inference, predominantly owing to unmeasured variables. Their method applies in settings where researchers have access to both RCT data, as well as observational data. They proceed by first providing a taxonomy of common causal biases, before demonstrating that, under a specific generative model, these different biases have unique "signatures" based on the sign of the covariance between the magnitude of the bias and the (conditional) variance of variables involved in the causal graph ($S$, the inclusion indicator; $A$, the treatment indicator; and $Y$, the outcome). The authors derive the theoretical signatures for these various biases, introduce a (strongly consistent) estimator for the covariance terms, and then demonstrate the proposed methods synthetic data and a real-world analysis.

**Compliance With Llm Reviewing Policy:**

Affirmed.

**Final Justification:**

As indicated in the discussion with the authors, following the rebuttal I have increased my appraisal of their work. The commentary in my initial review remains, though, the authors improved presentation markedly improves my assessment.

**Key Questions For Authors:**

**Q1. Characterization of the empirical results.**
Have I misinterpreted anything relating to your empirical results? Do you disagree with my framing that, using the technique, on the simulations presented, it appears as though there is very limited agreement with the correct type of bias from the method itself? Have I missed anything in reading through these results?

If I have missed anything, that would shift my concern from being tied to soundness, significance, and originality to one of communication.

**Q2. Methods for classification, and better classification metrics.**
Have you consider a process for actually taking the estimates that are produced and turning those into a suggestion as to what bias is present, if any at all? Have you looked at this at all beyond the hypothesis testing framework being employed here? Do you have any metrics on how accurate this actually seems to be, empirically?

At present, it is hard to assess the soundness of the method, and in particular claims to the soundness beyond the assumptions, without having a clearer picture of the actual performance here. Being able to resolve this would overcome many of the major issues identified.

**Q3. Sample size limitations**
Have you considered the performance of the techniques under smaller (more realistic) sample sizes? How does it scale? I would be interested to combine this with Q2 as well, to determine the scaling behavior of the methods, overall.

At the present, the apparent limited power is a severe limitation of the work. If this is not actually the case, it would be good to know, and would improve the significance substantially.

**Q4. Other modelling limitations**
Do you have any sense as to the performance of the techniques under the relaxation of other modelling components? Specifically, what happens if nuisance models are incorrectly specified? What happens if there are multiple forms of bias present? Etc. This type of robustness would be really great to understand, if this is to be made into a practical tool. Of course, in practice, it is highly unlikely that (in many settings) the full nuisance models will be able to be modeled correctly; if that fact undermines the method entirely, it is worth knowing, at a minimum.

Addressing questions relating to the robustness of the techniques would help establish soundness and significance.

**Limitations:**

The authors do directly indicate most of the major limitations of their work. However, there are two concerns with how this is done. First, they do not seem to acknowledge concerns relating to the power (namely, their simulations have incredibly large sample sizes with modest performance). Second, they seem to overstate the ability to relax the assumptions that form the basis of their most significance limitation. Based on the presented paper, I see limited evidence that these assumptions can be easily relaxed, and claims to the contrary serve to predominantly distort how limiting the limitations appear to be.

**Strengths And Weaknesses:**

## Soundness

### Strengths
- The proposed taxonomy accurately captures many causal biases, and describes them well in the context. These are well-motivated, accurate, and succinct descriptions. Even outside of the authors' proposed methods, this summary is sound and useful.
- Under the data generating assumptions, I see no fault with the derivations or proposed methods. I believe that the paper is mathematically sound.
### Weaknesses
* I am not convinced that there is sufficient evidence to suggest that the assumptions for the data generating mechanism are able to be relaxed, not without either further theoretical developments (that show why this is the case, or why this is the case in more relaxed settings) or further empirical evidence. As such, the claims of the paper are, for me, somewhat overstated: the classification of the types of biases is limited to the narrow set of scenarios that are considered.
* The simulations do not seem to provide evidence of a functioning method, as presented. Considering the proportion of "correct areas" reported in Figure 3, there are some settings that seem to be very concerning for accurate identification of the type of bias. For instance, the reported rates of agreement for "Selection Type I" appear to be, 62% on S, 4% on A, and 33% on Y. This has the correct contour (should be identified for S and Y, but not A), however, the actual rates of this are notably low. In the best case scenario, this means that using the results of this test correctly identified only 33% of these situations, but it could be the case that it never correctly identified the signature. Similar results occur for confounding, and to a lesser extent selection type II. Moreover, for the "no bias" case, there is something like 10% false positive rate, despite calibrating on $\alpha = 0.01$.
	* First, I think that it would be preferable to report the actual rate of correct identification, rather than the individual components; this would give a sense of, if a practitioner were using this, how frequently can they expect to correctly identify the rates.
	* If you do choose to consider the first point here, it is critical to determine how you want to do this. You can do the multiple testing and try to draw conclusions, though, I suspect that (based on the results) this is likely to be underpowered; perhaps there is someway to do joint classification, based on the hash table instead?
	* A secondary consideration here is for the values that are meant to be zero. If you are using standard significance testing, you cannot draw positive conclusions of zero equivalence here (just lack of evidence of non-zero values). This goes to the point in (2) regarding how to actually use these results to draw conclusions.
* To my mind, the concerns in point (2) are greatly amplified owing to the fact that the sample sizes in the simulation are incredibly large, both for the RCT and the observational data. In very many real-world settings sample sizes will be a fraction of this. Given that it appears that there are major power concerns even at the inflated sample sizes, this calls into question the efficiency (and as a result, utility) of what has been proposed, as it has been proposed. Further, given the current concerns in the simulations even for favorable settings (i.e., large sample sizes that follow the correct generative model), it is increasingly difficult to view the results as evidence of expanding beyond these assumptions.
	* Note: I do think it is plausible that a more refined classification mechanism could improve the detected power here; though, that would need to be justified and demonstrated.
* The assumption of a single bias being present also feels to me to be underexplored. I think that, even beyond the generating model, this is a major assumption that potentially induces concerns with the process. It is completely plausible that, if two biases were present, they would mimic the signature of a single bias. This can be explored directly, however, the concerns seem to mount when the signature that may be mimicked is the "no bias" signature.
## Presentation
### Strengths
* For the most part, the paper is well written, with minimal grammar/typesetting concerns.
* The background and methods are generally very well explained, and the motivation and language choices are appropriate and accessible.
* The overall structure of the manuscript works well.
### Weaknesses
* I do not think that the simulation results are presented in a way that is easily or accurately interpretable. I think that it is likely that a table does a mores succinct job at capturing the information presented in the figures, based on the interpretation that researchers are likely to desire here. For instance, some type of a confusion matrix, or similar, may be more appropriate based on what is being proposed.

## Significance

### Strengths
- It is an important and interesting question, to my knowledge distinct from existing lines of research, to attempt to classify the types of causal biases that are likely present in a given analysis. I think that the framing of this question, and some of the initial insights, are likely to be useful for practitioners, and have the capacity to be built on-top of going forward.
### Weaknesses
- The limitations arising from the data generating mechanism do greatly limit the direct significance of the work. Without theoretical support for why these seems "hashes" ought to hold  beyond these assumptions, or without a tremendous amount of empirical evidence that they do, it is not possible to read the results beyond the limited domain of applicability.
- The lack of power in actually identifying the true result further seems to minimize the significance of the result. With very large sample sizes, the use of the proposed hypothesis tests does not seem to have sufficient power to be directly useful. Without a method for working around this (e.g., a different method of classifying, a more powerful estimator, better empirical demonstration, etc.) the main result becomes the identification of the covariance hashes, under the specific data generating model; while this is interesting, it has limited significance.
## Originality

### Strengths
- To my knowledge, there has not been previous attempts to diagnose causal biases in this manner. The overall approach, and general contribution is, I believe, novel. The work also stands as distinct from structure-learning approaches.
### Weaknesses
- There are no major originality problems, save for concerns around whether the question that is claimed to be addressed in the manuscript is actually addressed; that is, classifying types of causal biases is novel, however, given the aforementioned limitations, it is not clear that this goal has actually been achieved, limiting the originality.

---

> ### Author Rebuttal · Authors · 2026-03-31
>
> We thank the reviewer for the careful and constructive review. We are glad that they found our contribution novel and the theory mathematically sound. We hear the central themes of the review: how the method should be interpreted empirically, how far the claims extend beyond the theoretical setting, and what the current limitations are in practice (*e.g.*, empirical power and consistency assumptions).
>
> We do our best to address these concerns, and we believe that our paper still offers value by taking a concrete first step on a difficult problem: moving from benchmarking average effect estimates to identifying *bias mechanisms*, with formal guarantees in an idealized setting and empirical evidence that the resulting signals remain informative beyond it.
>
> ## Reviewer Comments #1, #2, #3
>
> > “the simulations do not seem to provide evidence…”
>
> > “would be preferable to report the actual rate…”
>
> > “for the values that are meant to be zero…”
>
> This is a great point. We agree that Figure 3 does not present end-to-end classification performance in the clearest way. A joint summary such as a confusion matrix or hash-based classification table would be more informative, and we will add this in the revision.
>
> However, we would like to clarify a subtle point. In biased settings, the non-negative signals are coupled through the underlying bias magnitude: when a particular run happens to induce weaker bias, those are also the runs where each of the relevant covariance signals is more likely to be small and statistically insignificant. Thus, these errors are not independent failures, but often reflect the same low-signal instances. We will clarify this point and improve the presentation of the empirical results by reporting actual rates of correct identification, as suggested by the reviewer. We will also clarify that “zero” entries are currently operationalized as lack of evidence for a non-zero signal, rather than formal equivalence to zero.
>
> ## Reviewer Comments #4, #5
>
> > “there are major power concerns even at the inflated sample sizes”
>
> > “have you considered the performance under smaller sample sizes?”
>
> This is a fair point. We do not claim efficiency or optimality for the current plug-in covariance estimator. We do study sensitivity to sample size and covariate dimensionality in Appendix B.1 (Figure 8), and the trend is exactly what the reviewer points out: as the RCT becomes smaller, estimating the bias function becomes noisier and power deteriorates. We will surface this result more prominently.
>
> Importantly, however, we do observe that the covariate signals' powers (being significantly different from 0) across different bias scenarios compare to each other as predicted by the theory.
>
> ## Reviewer Comment #6, #7
>
> > “the classification of the types of biases is limited to the narrow set of scenarios…”
>
> > “what happens if there are multiple forms of bias present? … if nuisance models are incorrectly specified?”
>
> We agree that our sharp identification claims are limited to the setting analyzed in the theory, especially the single-bias setting and consistent nuisance estimation. We view that as an important first step toward a harder problem, namely identifying mechanisms rather than only benchmarking average treatment effects.
>
> At the same time, we do try to probe several departures from the theory empirically. Appendix B.1 studies alternative type-2 selection mechanisms (Figure 9), mixtures of multiple biases (Figure 10), and continuous rather than binary (U) (Figure 11). Section 6 and Appendix B.2 further examine the WHI case through targeted synthetic experiments that test our hypotheses for the observed signals there; see in particular the paragraph “Type 2 selection bias and transportability bias” and Figure 12.
>
> Consistency of nuisance function estimators is a common assumption that underlies many analytical results in the broader causal inference literature, particularly for plug-in and semi-parametric estimators, where reliable estimation of propensity scores or outcome models is needed to guarantee valid inference. In our setting, this assumption is especially important because inconsistency can introduce additional statistical bias that is difficult to disentangle from the causal bias signal itself. Even in modern approaches that relax this requirement via orthogonalization, some form of consistency, typically at weaker rates, remains essential [1]. We mention this in Table 1, but agree that the limitation deserves stronger emphasis.
>
> We thank the reviewer once again for their detailed review of our paper. We are happy to address any remaining questions that can help change your evaluation of our paper to a more positive one!
>
> [1] Chernozhukov, V., Chetverikov, D., Demirer, M., Duflo, E., Hansen, C., Newey, W., & Robins, J. (2018). Double/debiased machine learning for treatment and structural parameters.

---

> > ### Author Rebuttal · Reviewer_HgMk · 2026-03-31
> >
> > With the authors' willingness to update the presentation of certain results, clarify particular points, etc. that shifts my overall view. While I do think that certain concerns remain (which the authors do seem to acknowledge), overall this leaves it in a better state than after my initial review. My review will be updated to reflect this.

---

> > > ### Author Response · Authors · 2026-04-02
> > >
> > > We greatly appreciate the reviewer's engagement with our rebuttal and increasing their score. We will meticulously implement the promised updates.

---

### Official Review · Reviewer_oDUy · 2026-03-10

**Soundness:** 3
**Presentation:** 3
**Significance:** 2
**Originality:** 3
**Overall Recommendation:** 4
**Confidence:** 3

**Summary:**

This paper studied the identification of the underlying bias mechanism in observational studies. The paper proposed a new connection between the predictive performance of nuisance function estimators and the bias manitude. Moreoever,  using the covariance between the prediction error and magnitude of the bias, the paper proposed new methods to provably distinguish different bias mechanisms. Lastly, the paper developed consistent estimators of the covariance and conducted synthetic experiments and real-world analysis using data from Women’s Health Initiative.

**Compliance With Llm Reviewing Policy:**

Affirmed.

**Key Questions For Authors:**

- does the bias identification cover all types of possible biases?
- what is the efficiency of the covariance estimator? is that optimal?

**Limitations:**

yes.

**Strengths And Weaknesses:**

Strengths:
- Overall the paper is well-organized and presented. It addressed an important and practical problem in observational studies that the identification of the bias mecahnism is often difficult.
- In terms of bias mechanism identificaiton, the paper provided interesting new insights to use the covariance between the prediction error and magnitude of the bias to distinguish between diffirent types of biases. Estimating the covariance can be challenging in practice, but the one-to-one correpondence between the covariance and bias type provided some novel insights.
- the paper also provided a clear taxonomy on different bias mechanisms.

Weakenss:
- the set of assumptions are quite stringent. In particular the consistency of Nuisance Estimators is strictly required and cannot be relaxed.
- it is unclear if the identification still works where there exists a mixture of multiple types of bias. For example, In table 2, if all three covariances are positive, both confounding and Selection bias type 1 could exists. Do the theoretical guarantee and bias adjustments still hold?

---

> ### Author Rebuttal · Authors · 2026-03-30
>
> We thank the reviewer for their thoughtful feedback, and for highlighting the novelty of the insights provided and practical importance of the problem. Below, we address their questions and concerns.
>
> ### Reviewer Comment #1
>
> > the set of assumptions are quite stringent...
>
> We agree that our assumptions are relatively strong, but they stem from the more ambitious goal we tackle, which is not just estimating or benchmarking *average* treatment effects, but identifying underlying bias *mechanisms* via higher-order signals. In this setting, consistency is required to ensure that the covariance signals reflect causal bias rather than additional estimation error, which would otherwise be indistinguishable, as noted in Table 1. Moreover, consistency of nuisance function estimators is a common assumption that underlies many analytical results in the broader causal inference literature, particularly for plug-in and semi-parametric estimators where reliable estimation of propensity scores or outcome models is necessary to guarantee valid inference. Even in modern approaches that relax this requirement via orthogonalization, some form of consistency (at weaker rates) remains essential [1]
>
> More generally, the full set of assumptions enables sharp identification results that would not be attainable under weaker conditions. We view this as an important first step in a new direction, formalizing when such mechanistic insights are possible, and believe that systematically relaxing these assumptions, for example via less restrictive distributional assumptions, is both non-trivial and an important avenue for future work.
>
> ### Reviewer Comment #2
>
> > it is unclear if the identification still works where there exists a mixture of multiple types of bias...
>
> It is a significantly harder task to identify multiple sources of biases. Our theoretical results are derived under the assumption that a single type of bias is present, which we see as a critical and necessary first step. Nonetheless, due to the practical relevance of the multi-bias scenario, we extensively explore it empirically both in our real-world and synthetic experiments. At the end of section 6 (second to last paragraph), we discuss the implications of the existence of multiple biases on our statistics within our real-world study. Furthermore, in Appendix B.1, we conduct a set of synthetic experiments to investigate the behavior of our signals under various combinations of biases. It is an important future work to derive formal guarantees for the identification of a combination of bias types. However, it requires non-trivial effort and additional assumptions, and the current manuscript can serve as a basis for such analyses.
>
> ### Reviewer Comment #3
>
> > what is the efficiency of the covariance estimator? is that optimal?
>
> In our current formulation, we do not establish efficiency or optimality guarantees for the covariance estimator. The estimator is a plug-in object built from nuisance function estimates, so its asymptotic behavior is governed by the convergence rates of those components. Under consistency and standard conditions, it is consistent and asymptotically normal, but not necessarily efficient. Establishing semiparametric efficiency would require characterizing the efficient influence function for the covariance functional and constructing an orthogonal (e.g., cross-fitted) estimator that attains the efficiency bound. We view this as an interesting and non-trivial extension. Importantly, we note that our main contribution is the identification of covariance signals as signatures of the underlying bias mechanism, and the statistical properties we develop regarding their estimators are more of secondary contributions.
>
> We thank the reviewer once again for their detailed review of our paper. We are happy to address any remaining questions that can help change your evaluation of our paper to a more positive one!
>
> [1] Chernozhukov, V., Chetverikov, D., Demirer, M., Duflo, E., Hansen, C., Newey, W., & Robins, J. (2018). Double/debiased machine learning for treatment and structural parameters.

---

> > ### Author Rebuttal · Reviewer_oDUy · 2026-04-05
> >
> > The authors' reponse addressed my major questions.

---

> > > ### Author Response · Authors · 2026-04-05
> > >
> > > We greatly appreciate the reviewer's engagement with our rebuttal and are pleased to hear that all their major questions are addressed.

---

### Official Review · Reviewer_5AJK · 2026-03-12

**Soundness:** 3
**Presentation:** 2
**Significance:** 3
**Originality:** 2
**Overall Recommendation:** 3
**Confidence:** 4

**Summary:**

The paper provides methodology for diagnosing the type of bias present in an observational study, when a randomized controlled trial point of comparison is available. Through the estimation of the bias and nuisance probabilities, the paper gives a method to estimate covariance quantities between the conditional bias and conditional variance of three different variables, which can be jointly classified to determine which of four bias types. The paper gives plots to visualize the bias phenomena at hand and the ability of their estimators to identify the bias type.

**Compliance With Llm Reviewing Policy:**

Affirmed.

**Key Questions For Authors:**

1. Throughout the paper, the authors refer to “Theorem” 2.1, 4.1, etc., when the original statement instead is given as an “Assumption”, “Definition”, etc. It would be good to keep the terminology consistent for ease of reference.

2. I am confused why (1) and (2) define the CATE as the conditional expectation of Y^a. I interpret a CATE as a treatment effect and therefore a difference between Y^1 and Y^0, or similar.

3. The authors give plots with percentages for each covariance indicating how significantly away from 0 each is, but it is unclear how one should interpret the percentage. They imply that ~33% is significant, and that 10% is insignificant, but what is the right heuristic for interpreting these numbers?

**Limitations:**

Yes.

**Strengths And Weaknesses:**

Soundness: The submission is technically sound, with well-defined and well-argued theory and motivating examples. The empirical results adequately show the method’s benefits, with many further results in the appendix. The authors enumerate their method’s strengths and weaknesses, including a table with assumptions and whether they can be relaxed.

Presentation: The submission is generally well written, and the intuition for their methods is clear. The narrative is sometimes unclear, especially in the main section, which could be better structured to highlight the main contribution being the bias detection method.

Significance: The paper concerns the classification and detection of bias within an observational study, which is important for study design and analysis. The paper shows how one can interpret the results of their method, but it would be helpful to further detail how one could use the result of their method to improve decision making or estimation in practice.

Originality: This work develops a novel methodology for identifying the source of bias in an observational study. The taxonomy of biases appears to be more of a synthesis than an original contribution.

---

> ### Author Rebuttal · Authors · 2026-03-29
>
> We thank the reviewer for the thoughtful review and for recognizing the technical soundness of the paper, the clarity of the motivating examples, and the utility of the assumptions table. We also appreciate the constructive suggestions regarding presentation and practical significance.
>
> # Responses to Key Questions
>
> 1. Thank you for pointing out the notational inconsistency caused by our use of the LaTeX `cleveref` package. We will fix the erroneous references to Assumptions and Definitions that currently appear as Theorems.
>
> 2. This is a fair point. The standard definition of the CATE is indeed a contrast between two potential outcomes. In our development, we work directly with the conditional expectations of individual potential outcomes because this leads to a cleaner analysis of the bias function and its covariance structure. These quantities still have a causal interpretation since they are defined in terms of counterfactual outcomes. We will revise the paper to make this choice explicit and avoid confusion around the term CATE.
>
> 3. We understand why the percentages in the plots can be confusing. We do not interpret values such as 33% as “significant” and 10% as “insignificant.” Rather, these percentages report the fraction of experimental runs in which a given covariance was found to be statistically significantly different from zero under a given bias setting. The intended message is that these rates tend to align, imperfectly but meaningfully, with the qualitative patterns predicted by the theory. We will clarify this interpretation in the main paper. We will also consider adding a clearer joint summary of the three signals, since that is ultimately closer to how the method is used for diagnosis in practice.
>
> # Responses to Other Comments
>
> ## Reviewer Comment #1
>
> > “the narrative is sometimes unclear... the main section... could be better structured to highlight the main contribution”
>
> In the revision, we will restructure the main paper so that the bias detection framework and its practical interpretation are surfaced earlier and more clearly, while streamlining lower-level technical details where appropriate.
>
> ## Reviewer Comment #2
>
> > “it would be helpful to further detail how one could use the result of their method to improve decision making or estimation in practice”
>
> Our intention is that identifying a likely bias mechanism helps practitioners decide how to refine the study design or analysis. For example, evidence consistent with hidden confounding points toward improving covariate adjustment or collecting proxies, whereas evidence consistent with selection bias suggests revisiting inclusion criteria, follow-up mechanisms, or cohort construction. We will expand this practical discussion in the revision.
>
> ## Reviewer Comment #3
>
> > “The taxonomy of biases appears to be more of a synthesis than an original contribution”
>
> We agree that the taxonomy itself is partly a synthesis of known bias types. Our intended contribution is not the taxonomy in isolation, but the way it is formalized and connected to a new diagnostic methodology based on covariance patterns between estimated bias and nuisance-prediction uncertainty. We will sharpen this distinction in the revision.
>
> More broadly, we do not claim that our method provides an error-free procedure for identifying bias type, which is a very difficult problem. Our claim is more modest: we believe the paper offers a meaningful initial step toward understanding the mechanisms behind flaws in observational studies.
>
> We thank the reviewer once again for their detailed feedback! We are particularly encouraged by the 'Good' ratings for the paper’s Soundness and Significance. Given that the primary concerns raised were focused on terminology and presentation, which we attempted to address in our rebuttal, we hope these clarifications allow for a re-evaluation of the overall score to better align with the positive technical assessment. If there are any remaining technical concerns that led to the current score, we would be more than happy to address them further during the discussion phase!

---

> > ### Author Rebuttal · Reviewer_5AJK · 2026-04-03
> >
> > I thank the authors for the rebuttal. Their proposed changes should help the readability of the paper, and I will revise my presentation score to 3 – good, but maintain my overall evaluation. I’d like to highlight in question 3 that it would be helpful to have an explicit ballpark for what percentages indicate presence of bias, but it seems the authors will address this

---

> > > ### Author Response · Authors · 2026-04-03
> > >
> > > We thank the reviewer for engaging with our rebuttal. With respect to question 3, we would like to clarify that the percentages themselves do not indicate the presence or absence of bias. Instead, they are the "percentages" of 200 independent runs where the signal is statistically significant different from zero (*i.e.,* 200 different runs for each setting including those with "no bias" and with different bias mechanisms). The main takeaway is that those percentages align with the identification results for specific bias mechanisms presented in Table 2.
> > >
> > > While we regret that our response was not sufficient to change the overall rating, we are encouraged by the good ratings and remarks regarding the soundness, presentation, and significance of our work.

---

### Decision · Program_Chairs · 2026-04-30

**Decision:**

Accept (regular)

**Comment:**

This paper reveals that the mechanisms behind observed biases can be disclosed through the relationship between bias magnitude and the predictive performance of nuisance function estimators. The proposed mechanism identification framework is validated through extensive synthetic experiments and a real-world case study, showing its effectiveness in uncovering bias mechanism.

The idea presented in this paper is novel. The paper classifies various causal biases using causal graphs and develops theoretical foundations along with a framework for bias identification. The theory and experiments are sound. The paper is well written. The framework is potentially useful in understanding biases and in causal inference practice. Reviewers have raised a few issues regarding presentation, experimental design, result interpretation, and assumptions and practical limitations. Overall, reviewers are satisfied with the authors’ responses to their questions and concerns.

The authors should update the paper to incorporate the changes promised during the rebuttal process.